# Chemistry-intuitive explanation of graph neural networks for molecular property prediction with substructure masking

Zhenxing Wu[1,2], Jike Wang[1,2,3], Hongyan Du[1,2], Dejun Jiang [1,2], Yu Kang [1], Dan Li[1], Peichen Pan [1], Yafeng Deng[2], Dongsheng Cao [4] ✉, Chang-Yu Hsieh [1] ✉ & Tingjun Hou [1] ✉

Graph neural networks (GNNs) have been widely used in molecular property prediction, but explaining their black-box predictions is still a challenge. Most existing explanation methods for GNNs in chemistry focus on attributing model predictions to individual nodes, edges or fragments that are not necessarily derived from a chemically meaningful segmentation of molecules. To address this challenge, we propose a method named substructure mask explanation (SME). SME is based on well-established molecular segmentation methods and provides an interpretation that aligns with the understanding of chemists. We apply SME to elucidate how GNNs learn to predict aqueous solubility, genotoxicity, cardiotoxicity and blood–brain barrier permeation for small molecules. SME provides interpretation that is consistent with the understanding of chemists, alerts them to unreliable performance, and guides them in structural optimization for target properties. Hence, we believe that SME empowers chemists to confidently mine structure-activity relationship (SAR) from reliable GNNs through a transparent inspection on how GNNs pick up useful signals when learning from data.

In the past decades, deep learning (DL) has gained tractions and has been successfully applied to predict a broad range of molecular properties due to its strong capability to model complex nonlinear relationships between structures and properties[1–5]. However, this superior capability to capture intricate nonlinear relationships is originally achieved at the expense of model interpretability, leading to black-box predictions without insights[6–8]. For scientific investigations, such as drug discovery, what we want to know is not only the predictions of a model but also an explanation to validate scientific hypothesis and gain actionable insights to refine our investigations, such as hints for molecular structural optimization[7].

Ideally, an insightful interpretation should uncover potentially generalizable principles or actionable insights behind the structure-activity or structure-property relationships, thereby allowing scientists to gauge the reliability of model predictions. In fact, some researchers in medicinal chemistry might value the interpretability of a model over its accuracy[8–11] if a small sacrifice of accuracy can enhance its interpretability. Motivated by the reasons above, many explainable artificial intelligence (XAI) methods have been proposed to address the general lack of interpretability in molecular GNNs in order to augment human reasoning and decision-making[7,8,12,13].

However, most existing methods directly adopt attribution methods for XAI developed in other fields. Different from other unrelated applications, such as natural language processing and image recognition, in which DL has been shown to excel, there is no naturally applicable, complete, and 'raw' molecular representation. After all,

[1]Innovation Institute for Artificial Intelligence in Medicine of Zhejiang University, College of Pharmaceutical Sciences, Zhejiang University, Hangzhou 310058 Zhejiang, P.R. China. [2]CarbonSilicon AI Technology Co., Ltd, Hangzhou 310018 Zhejiang, P.R. China. [3]National Engineering Research Center for Multimedia Software, School of Computer Science, Wuhan University, Wuhan 430072 Hubei, P.R. China. [4]Xiangya School of Pharmaceutical Sciences, Central South University, Changsha 410004 Hunan, P.R. China. ✉e-mail: oriental-cds@163.com; kimhsieh2@gmail.com; tingjunhou@zju.edu.cn

molecules themselves are models that scientists conceive[8]. The choice of molecular representation could be a limiting factor for model explanation and performance because it inherently determines the kind of chemical information readily available for model training and prediction. A common choice is molecular graphs, which intuitively correspond to chemical structures (nodes to atoms, edges to chemical bonds, and subgraphs to substructures such as functional groups). This observation indicates molecular graphs as a suitable representation with great potential for chemist-friendly interpretability. Recently, several approaches have been proposed to analyze and interpret how GNNs make predictions[14–19]. However, most methods essentially attribute GNNs' predictions to individual nodes, edges, or node features. This kind of interpretability is only partially compatible with chemists' intuition at best. Chemists are more accustomed to comprehending the causal relationship between molecular structures and properties in terms of chemically meaningful substructures, such as functional groups, rather than individual atoms or bonds.

Following the survey work reported by Yuan et al.[19], existing explanation methods can be categorized into 5 classes: gradients/features-based methods, decomposition methods, surrogate methods, generation-based methods, and perturbation-based methods. Gradients/features-based methods employ gradient values or feature values to assess the importance of input graph nodes, edges or node features by extending existing image explanation techniques to the graph domain, such as sensitivity analysis (SA)[18], guided back-propagation (GBP)[18] and class activation mapping (CAM)[20]. Decomposition methods decompose the original model predictions into several terms following the backpropagation and take these terms as the importance scores of the corresponding input features (graph nodes or edges) to explain GNN models, and the representative works in this category include layer-wise relevance propagation (LRP)[18,21], excitation backpropagation (Excitation BP)[20] and GNN-LRP[22]. The above two types of methods use gradients and closely-related back-propagation to explain models, so they can only be used to explain a single GNN model and cannot be applied to study ensemble methods. Since many high-performing property prediction models are integrated from multiple sub-models, we have to rely on alternative methods to derive explanations. For instance, surrogate methods do not suffer from the above constraints. They employ simple and interpretable models such as linear models based on interpretable descriptors to capture local relationships of deep graph models around the input data, and the representative works include GraphLime[23] and PGM-Explainer[17]. The explanations of the prediction of a surrogate model are taken as the explanation for the original model. However, the surrogate methods usually do not work with molecular graphs and are not fully compatible with expert knowledge based on reasoning with molecular graphs. XGNN is the only generation-based method that renders high-level explanations of GNNs by generating graph patterns that support the given prediction[24] of an underlying GNN model. As the only global GNN interpretation method, it has made a good attempt, but the explainable graph patterns it generates are often meaningless chemical fragments and the explanation related to mutagenicity is not intuitive. There are also perturbation-based methods that identify key features by monitoring the degrees of changes in the predictions through perturbing different input features, such as GNNExplainer[15], PGExplainer[16] and SubgraphX[19]. GNNExplainer and PGExplainer provide subgraph-level explanations by combining nodes or edges to form subgraphs through post-processing, but the important nodes or edges highlighted by these methods are not guaranteed to be connected as one fragment. Sub-graphX can identify connected subgraphs with Monte Carlo tree search. However, due to the complexity of chemistry rules, the fragments uncovered by these perturbative methods are often not chemically meaningful and prone to generating confusing or even frustrating interpretations for chemists. For example, SubgraphX

splits a common functional group such as nitro into oxygen atoms and nitrogen atoms for separate masking, but a nitro group should be treated as a whole when deducing structure-property relationships. Moreover, similar to gradient-based methods, SubgraphX cannot be directly used to study a consensus model integrated by multiple sub-models, since it identifies subgraphs through Monte Carlo tree search and different subgraphs may be identified in different sub-models. Given a trained consensus GNN model and an input graph, different sub-models will identify different important subgraphs, so it is difficult to give a consistent and unified important subgraph for a consensus model. As summarized above, all existing methods could potentially be improved to better suit the needs of the chemistry community, and a more detailed account on all these methods can be found in reference[14].

In short, chemists prefer to investigate molecular properties and reactivity in terms of chemically meaningful fragments rather than individual atoms or bonds without a proper characterization of the atomic environment. For example, when medicinal chemists try to optimize the structure of a lead compound, they often consider the replacement of bioisosteres and functional groups. A toxophore commonly used for toxicity structural alerts is also defined in terms of a chemical group that is responsible for toxic effects rather than a simple atom or bond without context. Similar reasoning can be applied to appreciate the benefits of introducing the concept of pharmacophore in drug design. All in all, it is more natural to explore the molecular properties in terms of fragments/subgraphs rather than atoms/bonds without a proper context. However, no method has been developed to explain the predictions from GNNs by attributing predictions to fragments derived from chemical knowledge. For instance, one can segment a molecule by breaking chemical bonds to generate retrosynthetically feasible chemical substructures (BRICS)[25].

In this work, motivated by the lack of a proper XAI technique for analyzing molecular GNNs strictly in terms of chemically meaningful fragments, we propose a perturbation-based explanation method named Substructure-Mask Explanation (SME) that effectively identifies the most crucial set of substructures in a molecule that are responsible for a model's prediction. Three different methods for molecular fragmentation (i.e., BRICS substructures[25], Murcko substructures[26,27], and functional groups) are incorporated into SME in order to give chemists a high flexibility by using various fragment combinations (from different fragmentation methods) to dissect a molecule in a way that is most appropriate for the task at hand and render a transparent view on the structure-property relationships hypothesized by a molecular GNN.

Four examples (i.e., aqueous solubility, genotoxicity, cardiotoxicity, and blood-brain barrier permeation for small molecules) are used to illustrate how chemists can benefit from using SME to rationalize the predictions of a reliable GNN model for the five aforementioned applications. Different from the previous GNN interpretation methods, SME can be used to explain property prediction models at the level of chemical intuitive fragments rather than at the level of atoms or bonds and hence provide meaningful SAR information to help medicinal chemists in structural optimization and de novo design.

## Results

To verify the effectiveness of SME in explaining molecular property prediction models, four different consensus models (ESOL, Mutagenicity, hERG, and BBBP) are developed. As shown in Table 1, each model achieved excellent performance on the test set. The ESOL model achieves the performance of $R^2$ of 0.927, and the Mutagenicity, hERG, and BBBP models also achieve sufficiently high ROC-AUC of 0.901, 0.862, and 0.919, respectively.

To illustrate the benefits of working with an attribution analysis that conforms to chemist's working style, we further discuss five application scenarios in which SME can help us better

utilize DL models for molecular property predictions and drug design. Under each scenario, we recommend different fragmentation schemes to go along with SME for analysis. A succinct overview is provided below.

**Scenarios 1**. Mine the SAR for a specific molecule (i.e., local explanation): analyze the attributions of different substructures in a molecule, and it would be better to consider all fragmentation schemes (i.e., a combination of BRICS substructures, Murcko substructures and functional groups);

**Scenarios 2**. Identify the most positive/negative components of a specific molecule: obtain the combined fragments with the most positive/negative attribution through the combination of BRICS substructures and Murcko substructures;

**Scenarios 3**. Mine the SAR for the desired properties (i.e., global explanation) on a statistical basis: analyze the functional group attributions on the whole dataset;

**Scenarios 4**. Provide guidance for structural optimization: compare the average attributions of different functional groups.

**Scenarios 5**. Molecule generation for desired properties: recombination of BRICS substructures with SME attribution scores.

**Table 1 | The performance of the consensus models on the test sets**

| Model | Type | Metric | Performance |
|---|---|---|---|
| ESOL | regression | $R^2$ | 0.927 |
| Mutagenicity | classification | ROC-AUC | 0.901 |
| hERG | classification | ROC-AUC | 0.862 |
| BBBP | classification | ROC-AUC | 0.919 |

## SME attribution visualization of Aqueous solubility

Chemists have explored and broadly understood the effect of a large number of functional groups on water solubility. Hence, solubility is often used to validate model explanation methods and it provides a good setting for testing SME. By masking different substructures (BRICS substructures, Murcko substructures, and functional groups), SME can calculate the attributions of substructures to predictions to help us understand how the model predicts a particular molecule. For example, the visualization of the attributions for **compound 1** is shown in Fig. 1a. The SMILES of **compound 1** and the compounds used in the subsequent analysis can be seen in Supplementary Table 4. As shown in Fig. 1a, hydroxyl is beneficial to enhance the model's prediction of hydrophilicity, while methyl, isopropyl, and cyclohexyl are beneficial to enhance the model's prediction of hydrophobicity, demonstrating that the attributions of the substructures are consistent with the known chemical knowledge. It can also be found that the attributions of the alkyl groups to hydrophobicity become higher as the number of carbon atoms increases (cyclohexyl>isopropyl>methyl), which also agrees with our chemical knowledge. In addition, through the combination of the BRICS substructures and Murcko substructures, SME can find the most hydrophobic substructure (the alkyl group) and the most hydrophilic substructure (hydroxyl). Although the atom mask in **compound 1** can also help to understand the attribution of each atom to hydrophilicity and hydrophobicity, the substructure mask unambiguously reveals a more systematic trend from the data (and easier for chemists to generalize and relate to existing knowledge), such as illustrated in the earlier comment that we may draw the observation that increasing the number of carbon atoms can increase the hydrophobicity of the alkyl groups. As the complexity of compounds increases, the explanation of the substructure mask will be easier to capture the pattern/trend behind the data and to deduce more

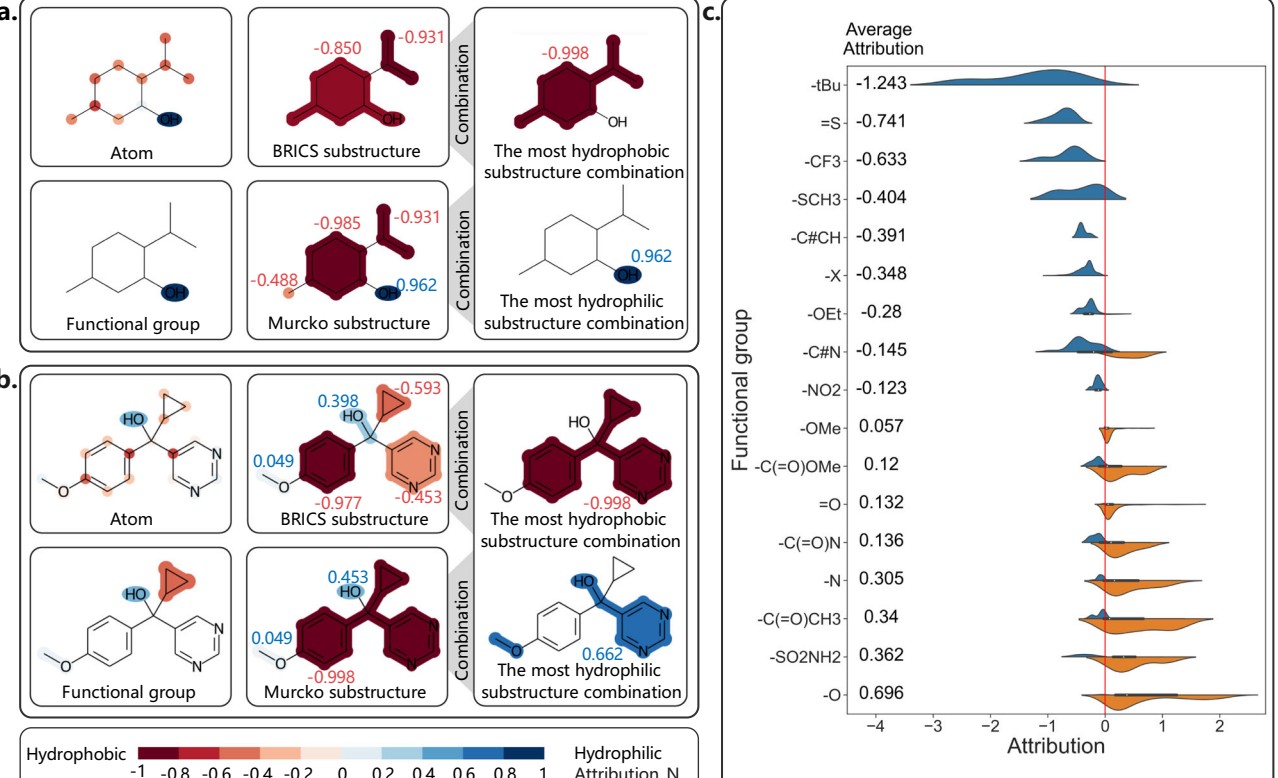

**Fig. 1 | The SME explanation of aqueous solubility. a** The attribution visualization of **compound 1**; **b** The attribution visualization of **compound 2**; **c** The attribution of the different functional groups in the whole ESOL dataset. Only the functional groups that appear more than ten times in the dataset are included in Fig. 1. Blue colors represent attributions lower than 0 and the substructure is favorable for hydrophilicity, while orange colors represent attributions higher than 0 and the substructure is unfavorable for hydrophilicity.

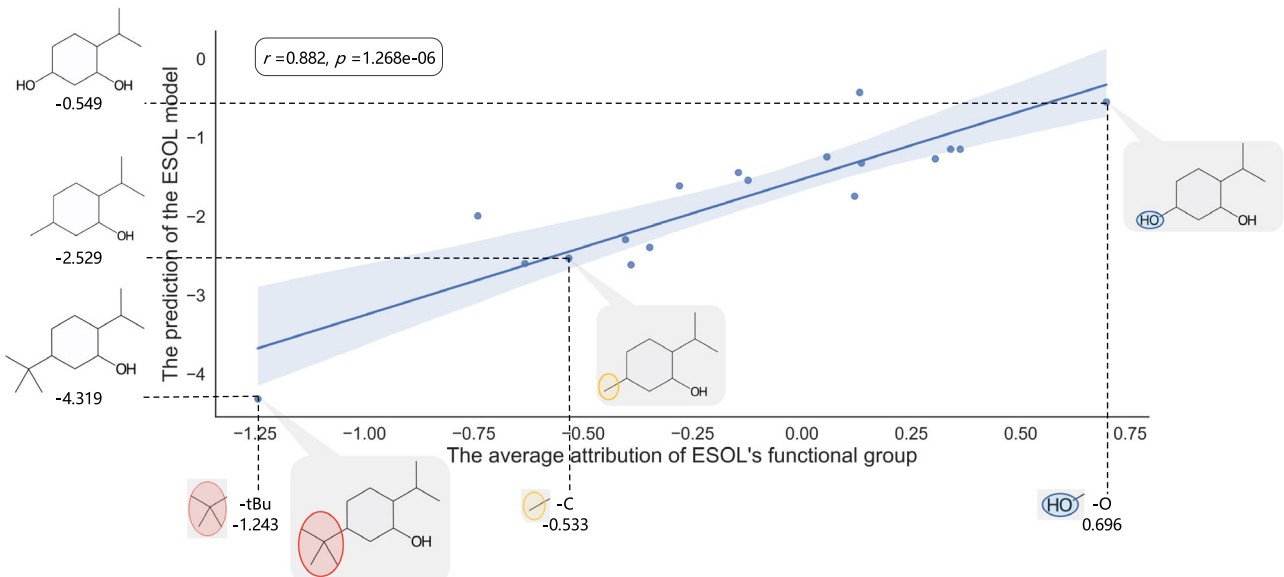

**Fig. 2 | The structural optimization of compound 1.** The source data can be found in Supplementary Data 1. The methyl group are changed to one of the functional groups in Fig. 1c and each point represents a compound. Yellow represents the initial functional groups, red represents a more hydrophobic functional group, and blue represents a more hydrophilic functional group. Regression line with the 95% confidence interval (the blue shading) and the spearman rank correlation test are used to describe the correlation between the hydrophilicity of molecules and the attribution of functional groups. The two-sided p-value indicates the significance of the correlation.

meaningful chemical knowledge than the simple atom mask. As shown in Fig. 1b, the explanation based on the substructure mask are obviously more intuitive than the atom mask. It is reasonable that cyclopropyl and phenyl are hydrophobic while hydroxymethyl and methoxy are hydrophilic (Fig. 1b, **BRICS substructure**).

## Functional group attribution analysis and structure optimization of aqueous solubility

As shown above, different substructures enable us to more precisely describe the boundary of fragments responsible for the predicted molecular property. Moreover, functional groups are predefined molecular substructures and can be used to evaluate the impact of the specific functional groups in the whole dataset. The attributions of different functional groups in the whole ESOL dataset are shown in Fig. 1c, and the functional groups with less than ten occurrences in the dataset are not considered.

As shown in Fig. 1c, SME tends to give positive attributions to common hydrophilic functional groups such as the hydroxyl (−O) and amino (−N) groups, and negative attributions to common hydrophobic groups such as the tert-butyl (-tBu) group. Hence, through the analysis of functional groups, SME can roughly assign the attribution of each functional group to the target property, and help to mine the SAR information.

Since the Superfast Traversal, Optimization, Novelty, Exploration, and Discovery (STONED) method enables rapid exploration of chemical space without a pre-trained generative model, some interpretability methods combined with STONED can guide structure optimization[7,28,29]. And the functional group attribution generated by SME can also aid in structural optimization. Taking **compound 1** as an example, the average attribution of different functional groups can be used to guide the optimization of its hydrophilicity. As shown in Fig. 1a, the Attribution_N of the methyl group is −0.488, and the attribution of the methyl group is −0.533. To structurally optimize **compound 1** to change its water solubility, we can manually change the methyl group to a functional group in Fig. 1c. The structural optimization result is shown in Fig. 2, demonstrating that the aqueous solubility of different compounds predicted by the model is highly correlated with the average attributions. The Spearman correlation coefficient between

the average attributions and predictions is 0.882 and the p-value is less than 0.05, indicating a significant positive correlation between the average attributions and predictions. Furthermore, by changing the methyl group to a functional group with a smaller average attribution, two of the three molecules are predicted to be less hydrophilic than **compound 1**, while changing the methyl group to a functional group with a larger average attribution, 13 of the 14 molecules are predicted to be more hydrophilic than **compound 1**. Hence, the above results demonstrate that SME does give a reasonable explanation. With the change of different functional groups, we can roughly sketch out the trend of the predictions by the model according to the average attributions, and then guide structural optimization.

## SME attribution visualization of mutagenicity

Mutagenicity is of great concern in drug development due to its hazards to human health. Some toxicophores and detoxifying groups are shown in Supplementary Fig. 1. The attributions of **compound 3** based on different types of substructures are shown in Fig. 3. A positive attribution means that the substructure is favorable for toxicity, and a negative attribution means that the substructure is considered detoxifying. As shown in Fig. 3, the nitro, amino, and quinone groups are beneficial to enhance the model's prediction of toxicity, while the carboxyl group is beneficial to enhance the model's prediction of non-toxicity. This is consistent with the existing literature reports that the aromatic nitro, aromatic amino, and quinone groups are identified toxicophores and the carboxyl group is an identified detoxifying group (Supplementary Fig. 1). In addition, through the combination of the BRICS substructures and Murcko substructures, SME can find the most toxic substructure (the aromatic nitro, aromatic amino, and quinone groups) and the most detoxifying substructure (the carboxyl group) as shown in Fig. 3.

## The combination can compensate for the flaws of SME in the classification problem

As shown in Fig. 3, compared with the regression problem (ESOL), the model assigns less attributions to fragments for the binary classification problem, and even only assigns an attribution of 0.003 to the

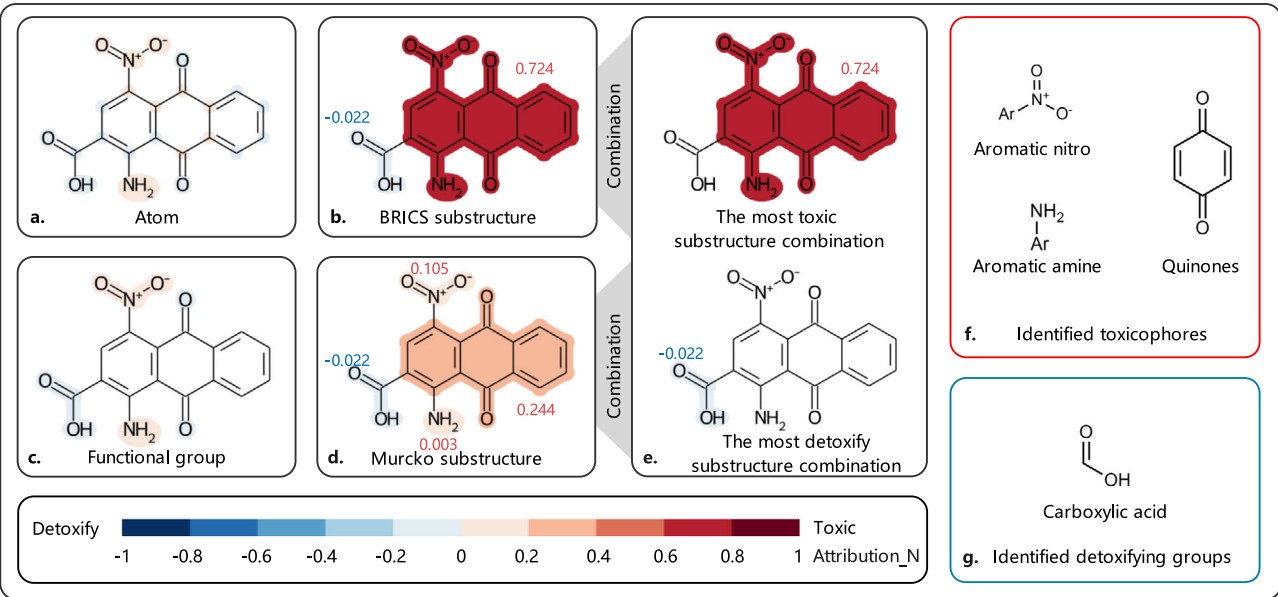

**Fig. 3 | The attribution visualization of compound 3. a** The SME explanation of **compound 3** based on atoms; **b** The SME explanation of **compound 3** based on BRICS substructures; **c** The SME explanation of **compound 3** based on functional groups; **d** The SME explanation of **compound 3** based on Murcko substructure; **e** The most positive/negative component of **compound 3**; **f** The identified toxicophores; **g** The identified detoxifying groups.

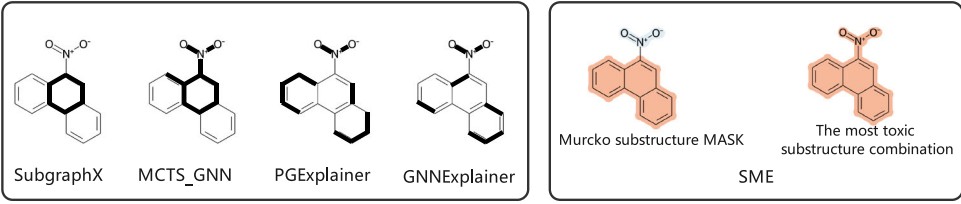

**Fig. 4 | The explanation results of compounds 7 for different methods.** The right side of the figure displays explanation results from SME, and the left side presents results from other methods.

amino group. The reason is that there are multiple toxic groups in some molecules, for example, there are three toxic groups (i.e., the aromatic nitro, aromatic amino and quinone groups) in **compound 3**. After removing a nitro group, the molecule still contains two toxic groups (aromatic nitro and quinone groups), so that the model still predicts the remaining part to be toxic. That is, $Y_{sub} \approx Y \approx 1$, which also causes SME to assign an attribution close to 0 to the toxic group. In short, this is why SME assigns an attribution close to 0 to the nitro group.

Compounds **4, 5,** and **6** are used to better illustrate the above flaws of SME. As shown in Supplementary Fig. 2, SME assigns the attributions close to 0 to the nitro and amino groups in **compounds 4** and **5**, but assigns a higher attribution value to the phenyl ring. The reason is that after removing the nitro and amino groups, the remaining part of the molecule still contains a toxic group, but when the phenyl ring is removed, the remaining part of the molecule no longer retains any identified toxicophores. This leads SME to judge that the phenyl ring is a very important toxic group, while the nitro and amino groups are not. The azide group is also an identified toxicophore. In **compound 6**, SME no longer judges the phenyl ring to be an important toxic group, and the reason is that the molecule still retains the azide group as a toxic group after the phenyl ring is removed. The above examples illustrate that SME will underestimate the attributions of toxicophore in molecules in the classification problem, especially when the molecule contains multiple toxic groups. Fortunately, by using our proposed method of taking fragment combinations for attribution analysis, we can largely fix the

observed flaws when only individual fragments are considered. As shown in Fig. 3 and Supplementary Fig. 2, the combination of fragments is always able to find all the toxic groups within the molecules tested in this study.

Moreover, we also compare the explanation results with the reported GNN explanation methods. The reported explanation results of **compound 7** are from Yuan's research and the explanation results for different explanation methods are shown in Fig. 4. Compared with the identified toxicophores in Supplementary Fig. 1, **compound 7** contains the toxicophores of aromatic nitro and polycyclic aromatic system. The reported methods have only identified a single benzene ring (SubgraphX and MCTS_GNN) or some discrete bonds (PGExplainer and GNNExplainer), and the results do not provide any help in understanding that polyaromatic rings are toxic fragments. Obviously, SME provides more reasonable explanation results than the reported methods through the identification of the attributions to more complete fragments rather than some discrete bonds. As shown in Fig. 4, SME can find all toxic substructures (aromatic nitro and polycyclic aromatic system) in **compound 7**.

## Functional group attribution analysis and structure optimization of Mutagenicity

The attributions of different functional groups in the whole Mutagenicity dataset are shown in Fig. 5 and the functional groups with less than ten occurrences are not considered. As shown in Fig. 5, the functional groups with the positive attributions are often the components of the identified toxic groups, such as nitroso ($-N=O$), nitro (-NO_2) and amino (-N). In addition, the functional groups with the

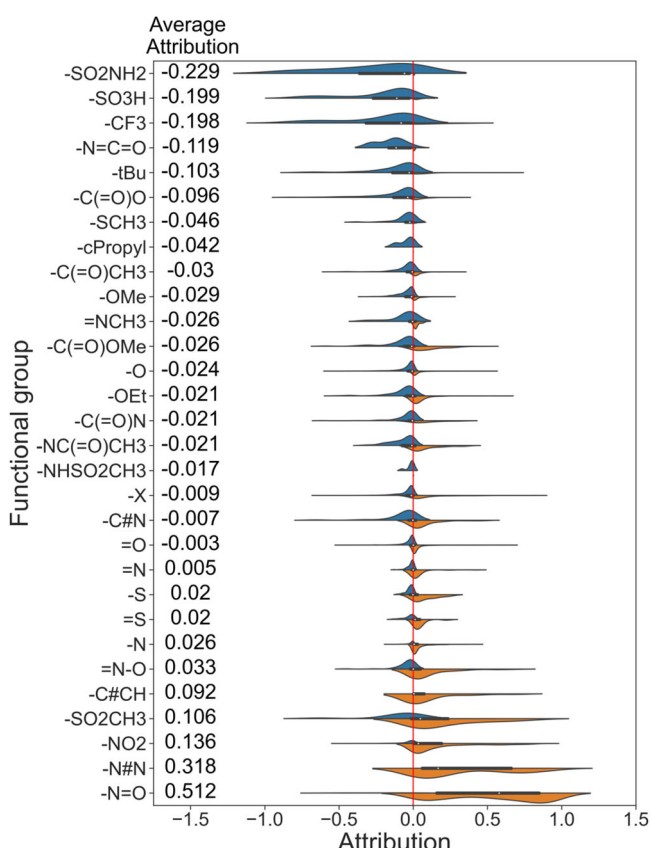

**Fig. 5 | The attributions of different functional groups in the whole Mutagenicity dataset.** Blue colors represent attributions lower than 0, while orange colors represent attributions higher than 0.

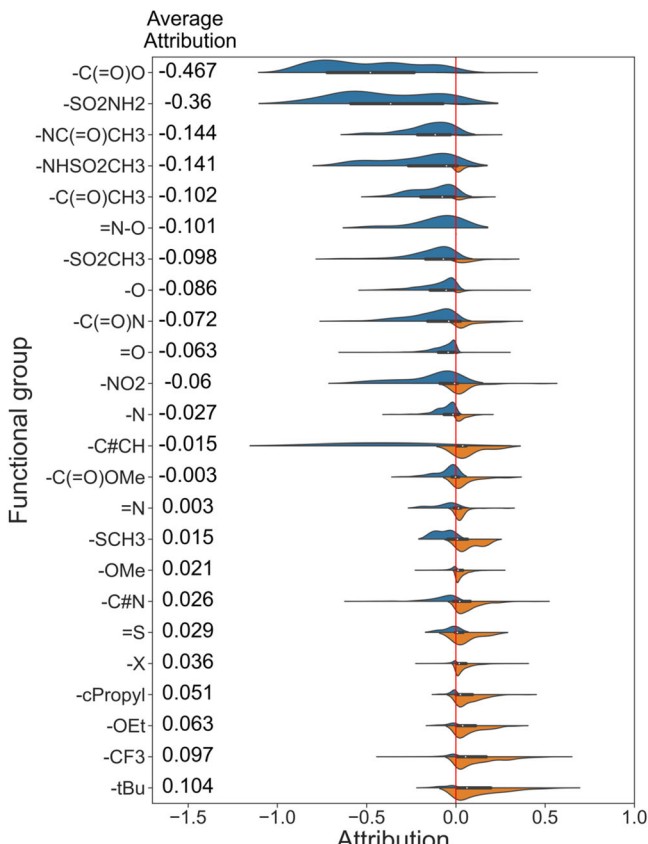

**Fig. 6 | The attributions of different functional groups in the whole hERG dataset.** Blue colors represent attributions lower than 0 and the substructure is unfavorable to induce toxicity, while orange colors represent attributions higher than 0 and the substructure is conductive to inducing toxicity.

negative attributions, such as sulfonamide ($-SO_2NH_2$), sulfonyl hydroxide (-SO3H), and trifluoromethyl (-CF3), are identified to be detoxifying groups. The above results show that SME can indeed extract reasonable SAR information from the model.

However, as shown in Supplementary Fig. 1, the isocyanate group is identified as a toxicophore, but SME always assigns negative attribution to the isocyanate group. And we explore what causes SME to make judgments contrary to the expert knowledge. There are only ten molecules in the dataset containing such functional groups, nine in the training set and one in the test set. In addition, only two of the nine molecules in the training set are toxic, while seven are non-toxic. The small amount of data and highly imbalanced labels cause the model to misjudge the isocyanate group as a detoxifying group. Moreover, the model also incorrectly predicts the only positive molecule in the test set that contains this group as a non-toxic molecule. Combining the SME explanation results and expert knowledge, we are able to discover problems that the model has learned the wrong information, and then provide directions to optimize the model. In other words, careful analysis on SME explanations can help us to judge if the model's predictions for the molecules containing certain structures are unreliable, and more similar molecules should be added to the training set to further improve the model's ability to predict such molecules as done in an active learning loop to improve model reliability with small amount of highly relevant data. For example, to optimize the prediction ability of the mutagenicity model for molecules containing the isocyanate group, it is necessary to add molecules containing the isocyanate group to the training set and provide more balanced data labels.

We also evaluate whether SME could guide the structural optimization of Mutagenicity, and the results are shown in Supplementary

Fig. 2. We sequentially replace the toxic functional groups in **compounds 4, 5,** and **6** (the amino, nitro, and isocyanate groups) with detoxifying groups (sulfonyl hydroxide, sulfonamide, and trifluoromethyl). As shown in Supplementary Fig. 2, after structural optimization, the prediction results of the model all change from close to 1 to close to 0, suggesting that SME really reflects how the model makes predictions.

## SME can reveal the hERG optimization strategies
Different from mutagenicity, there are no clearly identified toxicophores for hERG toxicity. Fortunately, in previous works, medicinal chemists have summarized a large number of strategies to reduce hERG toxicity[30–33]. Therefore, we explore whether SME can rediscover these optimization strategies to validate SME. The attributions of functional groups for hERG toxicity are shown in Fig. 6, and a positive attribution means that the substructure is favorable to toxicity while a negative attribution means that the substructure is unfavorable to toxicity. Introducing the hydroxyl ($-O$) groups and acidic groups ($-C(=O)O$) are common optimization strategies to reduce hERG toxicity, and SME also considers that the hydroxyl and acidic groups are unfavorable to hERG toxicity as shown in Fig. 6. Reducing lipophilicity is also a commonly used strategy to reduce hERG toxicity[32,33]. To verify whether SME mine this SAR, we evaluate the correlation of the hERG's functional group attributions and the ESOL's functional group attributions. As shown in Fig. 7, they are highly negatively correlated with a spearman correlation coefficient of $-0.86$ and a p-value of less than 0.05, indicating a high correlation between hERG toxicity and lipophilicity.

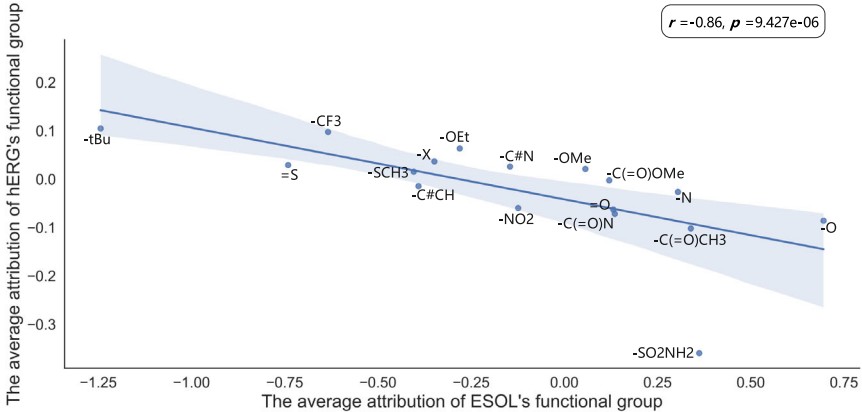

**Fig. 7 | The correlation of the average attributions between hERG and ESOL's functional groups.** The source data can be found in Supplementary Data 2. The regression line with the 95% confidence interval (the blue shading) and the spearman rank correlation test are used to describe the correlation of the average attributions; The two-sided $p$-value indicates the significance of the correlation.

## Examples of structural optimization for hERG toxicity in the real world

In the above structural optimization, the examples are consistent with the model prediction results. Changing functional groups with desired attributions can yield molecules with desired properties predicted by models, as exemplified in Fig. 2 and Supplementary Fig. 2. Here we find some real-world structural optimization examples for hERG toxicity to evaluate whether they meet the guidance provided by the average attributions derived from SME. Since a positive attribution means that the substructure is potentially more toxic while a negative attribution means that the substructure is 'detoxifying', SME can naturally provide three structural optimization strategies to reduce hERG toxicity as follows: 1. adding a functional group with an average attribution <0; 2. removing a functional group with an average attribution >0; 3. replacing a functional group with a more negative attribution. Some real-world structural optimization examples confirm the rationality of the above structural optimization strategies.

As shown in Fig. 8a, adding a carboxyl group with the average attribution of −0.465 to **compound 8**, the hERG toxicity of the compound decreases ($IC_{50}$ from 0.8 μM to 76.8 μM)[34]. And adding a amide group with average attribution of −0.069 to **compound 8** can also reduce the hERG toxicity of the compound ($IC_{50}$ from 0.8 μM to 17.1 μM)[34]. The above examples can demonstrate the effectiveness of adding a functional group with an average attribution <0 to reduce hERG toxicity. The second structural optimization strategy is to remove a functional group with an average attribution >0 to reduce hERG toxicity. As shown in Fig. 8b, removing a methoxy group with an average attribution of 0.021 to **compound 11** can reduce the hERG toxicity of the compound ($IC_{50}$ from 0.35 μM to 0.76 μM)[35]. Removing a chlorinated group with an average attribution of 0.036 can reduce the hERG toxicity of the compound ($IC_{50}$ from 0.85 μM to 5.49 μM)[35]. Another structural optimization strategy is replacing a functional group with a more negative attribution to reduce hERG toxicity. As shown in Fig. 8a, by replacing the amide group in **compound 10** with a more negative carboxylic group to obtain **compound 9**, the hERG toxicity is significantly reduced ($IC_{50}$ from 17.1 μM to 76.8 μM). Moreover, replacing the cyano group with the methylsulphonyl group (having a more negative attribution) reduces hERG toxicity[36]. As shown in Fig. 8c, replacing the amino group with the methylamide group (having a more negative attribution) also reduces the hERG toxicity ($IC_{50}$ from 0.12 μM to 2.6 μM)[37]. In addition, Rao et al. proposed a subjective method to evaluate the explanatory power of different interpretability methods and provide some hERG cliff molecular pairs. Some hERG cliff molecular pairs that did not appear in our training set also verify the effectiveness of SME in guiding structural

optimization, and these results are shown in Supplementary Fig. 3[38]. All in all, the above real-world examples can demonstrate that SME could guide the structural optimization of hERG toxicity.

## Attribution-based molecular generation for desired properties

The generation of molecules with desired properties has attracted the attention of computational chemists and a lot of excellent works based on DL has emerged in recent years. Through the BRICS fragments endowed with the attributions predicted by SME, we provide a molecular generation method that does not require any conditional generative model to generate molecules with target properties. BRICS fragments are originally designed to be recombined for molecular generation. Since SME can assign the attributions of desired properties to different BRICS fragments, it is natural to recombine the BRICS fragments with desired properties to generate molecules with desired properties.

The "positive BRICS fragments" (attribution>0) and "negative BRICS fragments" (attribution<0) are recombined to yield 3000 molecules, and the desired properties of these molecules are predicted. For different properties, the positive fragments include hydrophilic BRICS fragments, Mutag-toxic BRICS fragments and hERG-toxic BRICS fragments, while the negative fragments include hydrophobic BRICS fragments, Non-Mutag-toxic BRICS fragments and Non-hERG-toxic BRICS fragments. As shown in Fig. 9a, the distribution of the desired properties of the molecules generated by the recombination of the fragments with the desired attributions is indeed as expected. However, it can also be seen that there is still a partial intersection between the two classes of generated molecules, and even some molecules composed entirely of so-called "non-toxic fragments" are still predicted to be toxic. The main reason is that the attributions given by SME only represents that of the fragments in the current molecules, and their attributions may even be completely opposite in the new molecules. As shown in Fig. 5, the same fragment may also exhibit different attributions in different molecules. A simple solution to this problem is to sample fragments that are more likely to exhibit negative attributions throughout the dataset, since existing data suggests that these fragments tend to be non-toxic in the vast majority of existing molecules. And these fragments will also more easily retain their non-toxic properties in new molecules, such as sulfonyl hydroxide (-SO3H) and trifluoromethyl (-CF3) for mutagenicity toxicity.

Hence, in order to reduce the overlap between the two classes of generated molecules, we only take the top 20% fragments with the smallest attributions less than 0 as the negative fragments, and only

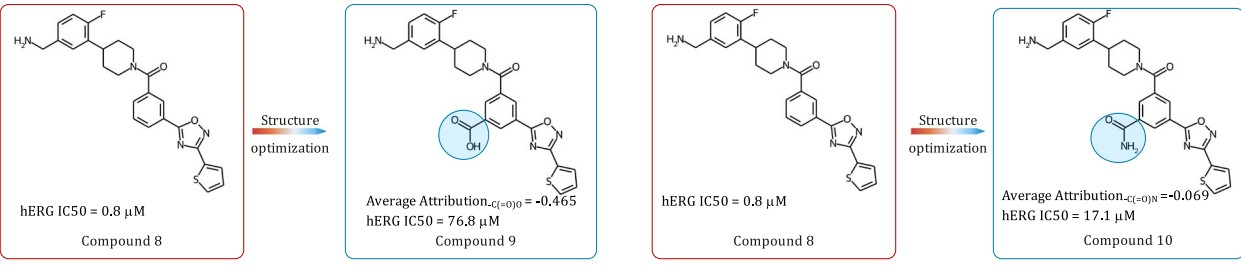

**a.** Add a functional group with an average attribution < 0 to reduce the hERG toxicity.

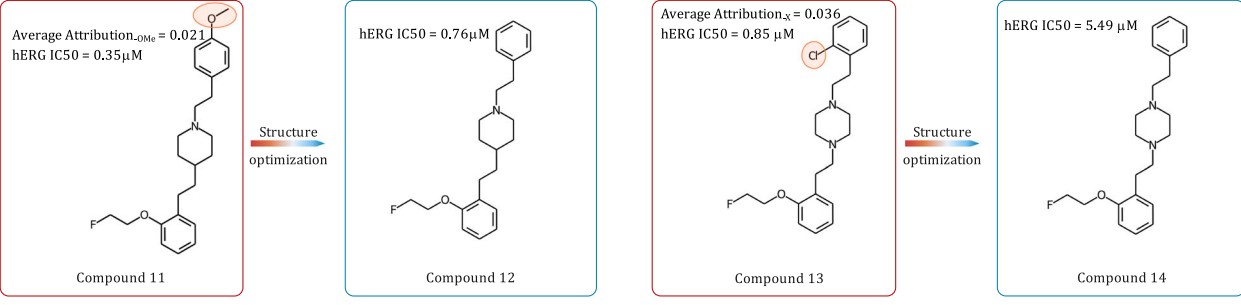

**b.** Remove a functional group with an average attribution > 0 to reduce the hERG toxicity.

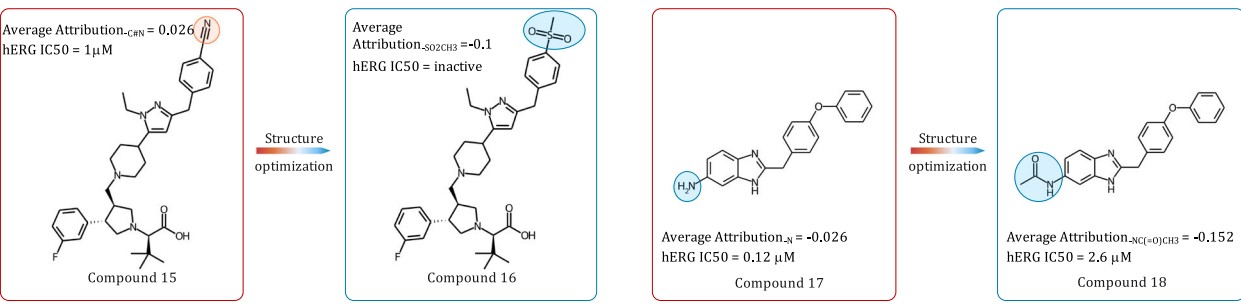

**c.** Replace a functional group with a more negative attribution to reduce the hERG toxicity.

**Fig. 8 | Some real-world structural optimization examples of hERG toxicity.**
**a** Add a functional group with an average attribution <0 to reduce the hERG toxicity; **b** Remove a functional group with an average attribution > 0 to reduce the

hERG toxicity; **c** Replace a functional group with a more negative attribution to reduce the hERG toxicity.

the top 20% fragments with the largest attributions higher than 0 as the positive fragments. A total of 3000 negative molecules and 3000 positive molecules are generated for 3 different tasks, respectively, and the corresponding properties are predicted. As shown in Fig. 9b, the overlap between the two classes of generated molecules is significantly reduced, demonstrating the effectiveness of the above strategy.

In addition, we also explore the application of attribution-based molecular generation for multiple desired properties. We merge the fragments co-existing in the top 20% Non-Mutag-toxic and Non-hERG-toxic fragments to obtain the Non-Mutag-toxic&Non-hERG-toxic fragments. 3000 molecules are generated from the Non-Mutag-toxic&Non-hERG-toxic fragments, and the predictions of the 3000 molecules from the Mutagenicity model and the hERG model can be seen in Fig. 9c. As shown in Fig. 9c, the vast majority of the 3000 generated molecules are predicted to be non-toxic, illustrating the potential of the attribution-based molecular generation approach in generating molecules with desired properties.

**How can SME establish SAR information with deeper explanations**

In this section, we show that SME (with its enforced attribution to chemically meaningful fragments) allows one to more easily establish a deeper understanding on why certain fragments make the molecule

more ideal for the target property. This deeper understanding is achieved by connecting fragments and some characteristic physiochemical properties that is conceptually known to be relevant to the property of interest, for instance, BBBP in this case.

Through the analysis of functional group attribution generated by SME, we can construct the desired relationship between molecular functional groups and the BBBP as shown in above Fig. 10a. Similarly, by constructing different prediction models of fundamental physicochemical properties, we can construct the desired relationship between molecular functional groups and the fundamental physicochemical properties. Subsequently, we then establish the relationship between BBBP and the fundamental physicochemical properties, providing a complementary perspective on what molecular properties (substructure or physiochemical attributions) hold the key to having the desired properties. Four fundamental physicochemical properties include molecular weight (MW), topological polar surface area (TPSA), lipid solubility (LogP) and the number of hydrogen bond donor (HBDs) calculated by RDKIT package are used as labels to construct different prediction models in this case. The overall process is shown in Fig. 10a and the detailed result can be seen in Fig. 10b–e. The performance of fundamental physicochemical property models can be seen in Supplementary Table 1. As shown in Fig. 10b–e, SME does not find a definite correlation between

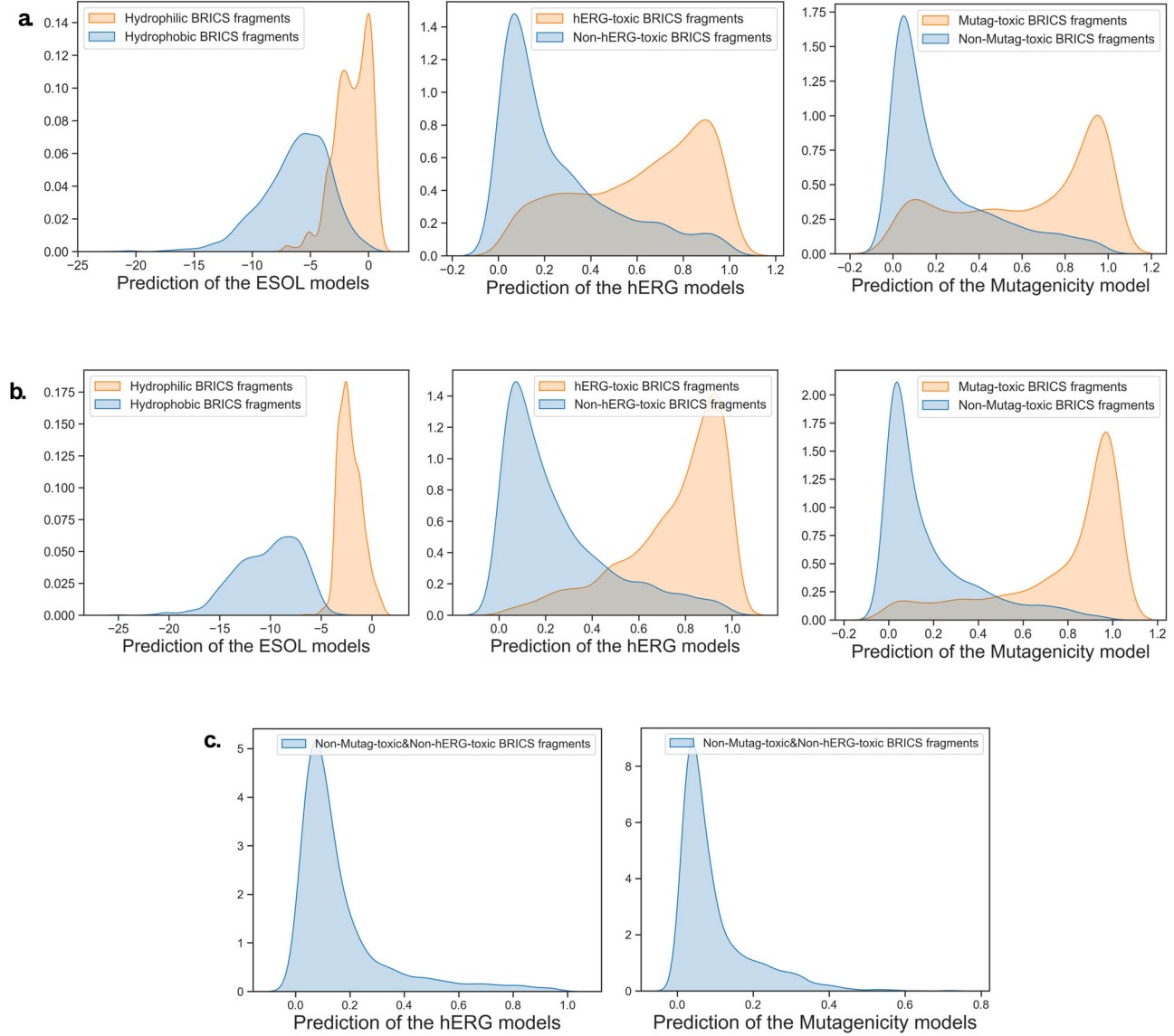

**Fig. 9 | Attribution-based molecule generation for desired properties. a** The molecules generated by recombination of BRICS fragments; **b** The molecules generated by recombination of top 20% BRICS fragments; **c** The molecules generated by recombination of top 20% Non-Mutag-toxic&Non-hERG-toxic BRICS fragments.

BBBP and MW. Moreover, SME finds that BBBP is positively correlated with LogP and negatively correlated with HBDs and TPSA, which is consistent with the results reported in previous studies. The above results demonstrate that chemists can mine SAR information through SME and use this valuable information to further expose favorable physiochemical properties these molecules must possess. Furthermore, converting information from favorable molecular substructures to favorable physiochemical/ biochemical properties by SME is an abstraction that may actually inspire further understanding of complex phenomena as done by Mittal, A. et al.[39].

## Discussion

In this study, we propose an intuitive SME to explain GNN models for molecular property predictions. Combining different well-designed molecular fragmentation methods (BRICS substructures, Murcko substructures, and functional groups), SME provides intuitive and chemistry-compatible explanations for all four tasks considered in this study. By analyzing the attributions of functional groups assigned by SME across the entire dataset,

we are able to explore how functional groups affect model predictions and infer structure-activity relationships. More importantly, the attribution analysis of functional groups can also provide guidance for structural optimization, and the experimental results and real-world examples confirm the validity and rationality of the attribution-based guidance. In addition, SME can also help to diagnose problems affecting a model's reliability and provide directions to further improve the predictive model. For example, the attribution of the isocyanate group assigned by SME based on the consensus mutagenicity prediction model is inconsistent with the existing expert knowledge, and we could easily identify the problem that the training data containing such functional groups is insufficient and unbalanced which may cause the model being unable to make correct predictions for molecules containing this functional group. Moreover, the recombination of the BRICS fragments with the attributions assigned by SME can be used to generate molecules with desired properties, which provides a way to efficiently generate molecules with desired properties that does not require training a generative model.

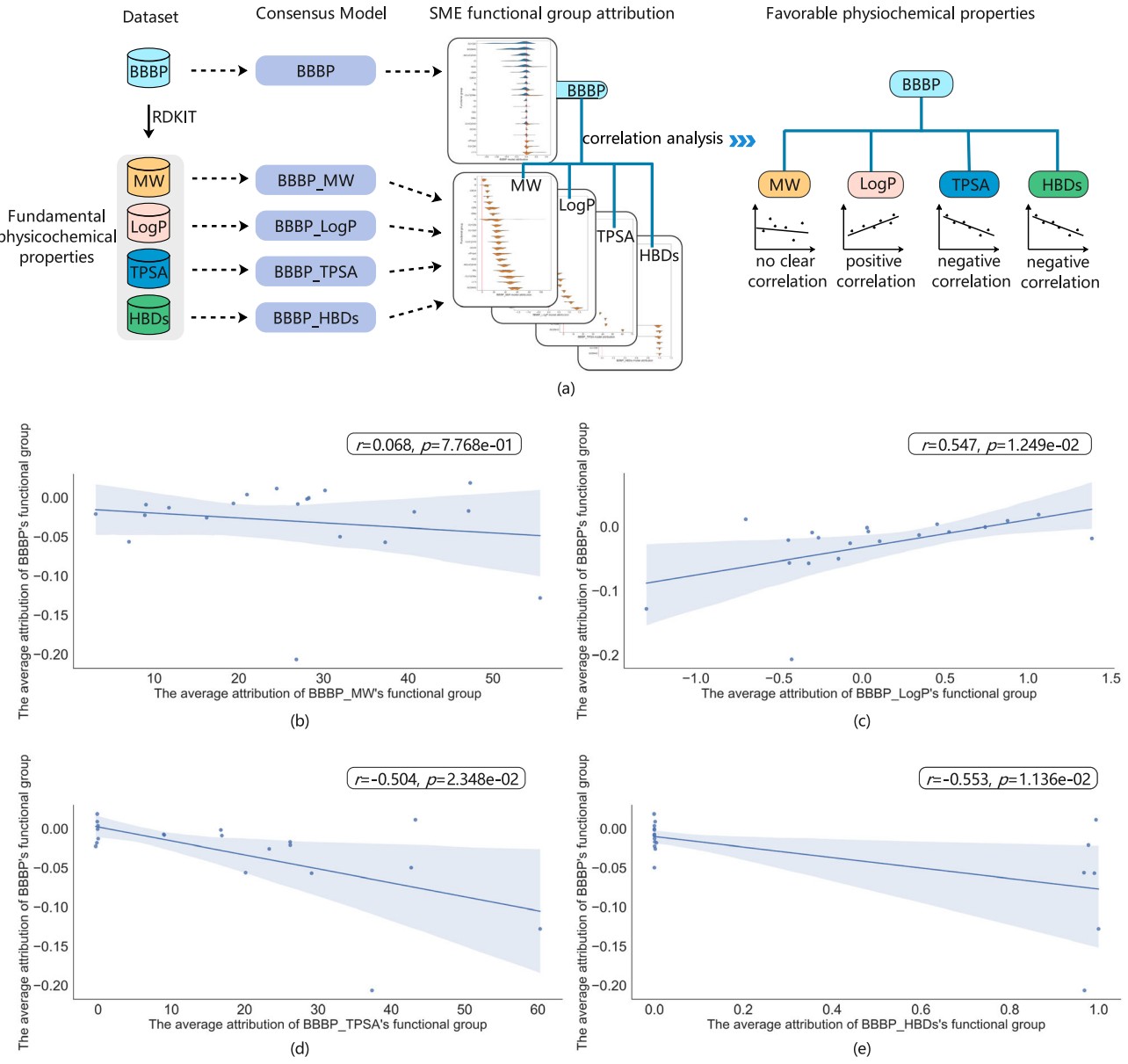

**Fig. 10 | The SAR information of BBBP mined by SME.** The source data can be found in Supplementary Data 3. **a** The flow chart of using SME to mine SAR information. The analysis of functional group attribution generated by SME are used to construct a desired relationship between molecular functional groups and the BBBP. By constructing different prediction models of physicochemical properties, the relationship linking molecular functional groups and physicochemical properties can be obtained based on the functional group attribution. With the functional group as a link, the relationship between BBBP and the physicochemical properties can be established. Regression lines with the 95% confidence interval (the blue shading) and spearman rank correlation test are used to describe the correlation between BBBP and the physicochemical properties. The two-sided p-value indicates the significance of the correlation. **b** The correlation of the average attributions between BBBP and BBBP_MW's functional groups; **c** The correlation of the average attributions between BBBP and BBBP_LogP's functional groups; **d** The correlation of the average attributions between BBBP and BBBP_TPSA' functional groups; **e** The correlation of the average attributions between BBBP and BBBP_HBDs's functional groups.

Just like all other explanation methods for deep learning models, SME also possesses several limitations. First, since SME (and other GNN-based interpretation methods) aims to explain what the GNN model has learned from the data, it is difficult for SME to mine reasonable SAR information when the GNN model has not learned the true or complete causality due to data paucity or bias. As a noteworthy example, tautomers present a challenge in the pursuit of accurate molecular property predictions and, accordingly, has implications for the effectiveness of SME. Second, as chemically intuitive explanations are enforced, intrinsically chemically meaningless patterns learned from data artifacts may remain concealed. Lastly, the current version of SME only supports the three substructures of BRICS, Murcko, and

functional groups, and does not allow the assessment of some other substructures such as bioisosteres. Irrespective of these limitations (some can be easily overcome such as the last point mentioned above), SME provides intuitive and insightful interpretability for chemists based on chemically meaningful substructures. As most existing attribution methods for molecular GNNs in chemistry are directly borrowed from other fields and lacks the kind of chemical intuition that SME insists by design, we believe its benefits outweigh its drawbacks. Particularly, we demonstrate how SME help chemists in 4 key druggability properties tasks (ESOL, Mutagenicity, hERG and BBBP). In sum, SME is a tool with great potential to assist chemists to gain an intuitive understanding of why a GNN model makes a certain

prediction, relating structural and physiochemical properties when it is appropriate to do so, and mine SAR information for structure optimization and de novo design.

## Methods

### Datasets

In order to verify the effectiveness of SME in explaining the predictions from a GNN model, we conduct experiments on four different molecular property datasets, concerning aqueous solubility (ESOL), mutagenicity, hERG-related cardiotoxicity and blood-brain barrier permeation (BBBP). We briefly give a short introduction to these four properties.

Aqueous solubility is essential for drug candidates and is one of the key physical properties of interest for medicinal chemists[40]. The flux of drugs across the intestinal membrane is proportional to the concentration gradient between the intestinal lumen and blood, so drugs with poor water solubility may result in poor absorption even if the permeation rate is high[40]. Therefore, accurate prediction of aqueous solubility will help optimize the absorption properties of drug candidates, and a large number of aqueous solubility prediction models based on ML have been reported[41–44]. Although accurate prediction of water solubility remains a difficult problem, chemists have explored and broadly understood the effect of a large number of functional groups on water solubility[42]. Hence, solubility is often used to validate model explanation methods and it provides a good setting for testing SME[7,45].

Mutagenicity reflects the ability of a compound to induce mutations in DNA. Many computational approaches have been developed in recent years to predict the mutagenicity of compounds in the early stage of drug development[46–50]. A large number of toxicophores and detoxifying groups have been identified and reported, and hence mutagenicity constitutes another popular setup for verifying whether ML models capture essential structure-property relationships to support their predictions[46,48,51–53].

Cardiotoxicity is another task that we consider, a highly crucial factor in drug design. Undesired hERG-related cardiotoxicity is one of the main reasons for drug candidate failures and drug withdrawal from the market[54,55]. And a number of hERG inhibition prediction models have been developed in recent years[46,56–58]. We will use this important property to illustrate how a chemistry-compatible explanation method, such as the proposed SME, not only rationalizing how a model makes predictions but also provides actionable insights to help us improve the design of drug candidates. As repeatedly implied in the introduction, a good explanation method should offer more than just a verification tool.

BBBP is another classic problem in computational chemistry. Since brain drugs need to penetrate the blood-brain barrier, the BBB permeability has been investigated in detail and many BBBP prediction models have been developed in recent years[59–61]. In this study, we consider BBBP datasets as an example to illustrate how one can further push the interpretation method discussed so far to generate a deeper explanation. As discussed later, not only will we highlight important fragments responsible for a molecule's BBBP, but also provide a rationale on how the responsible fragment determines some critical physiochemical properties of a molecule that subsequently affects its BBBP.

The detail on these four datasets is summarized in Supplementary Table 2 and further detail on the data curation can be found in Supplementary Table 3. In addition, the canonical SMILES of the compounds for analysis are shown in Supplementary Table 4.

### Substructure mask explanation

Under many real-world settings for molecular property predictions, a common practice is to integrate the predictions from multiple models. Thus, in this study, we first build 10 relational graph convolution

networks (RGCN) models based on different random seeds for initialization and then construct the consensus model by integrating the 10 RGCN models. The mean of the ten individual predictions will be used as the output of the final consensus model.

In recent years, GNN has attracted increasing attention in molecular property predictions due to its strong predictive power on graph-structured data. All GNN variants, such as graph convolution neural networks (GCN)[62], graph attention networks (GAT)[63], attentive FP[64], and directed message-passing neural networks (D-MPNN)[3], can be regarded as the special cases of differentiable message-passing frameworks. RGCN is an extension of the standard GCN by introducing edge features to enrich the messages used to update the hidden states in the network and has achieved excellent performance in link predictions and entity classification[65]. The propagation rule for each node $v$ is calculated via

$$h_v^{(l+1)} = \sigma \left( \sum_{r \in R} \sum_{u \in N_v^r} W_r^{(l)} h_u^{(l)} + W_0^{(l)} h_v^{(l)} \right) \quad (1)$$

where $h_v^{(l+1)}$ is the state vector of target node $v$ after $l+1$ iterations, $N_v^r$ denotes the neighbors of node $v$ under the relation (edge) $r \in R$ and $R$ denotes the set of edge types. The neighbors of node $v$ refers to the nodes directly connected to node $v$. $\sigma()$ is an element-wise activation function, and ReLU() is adopted in this study. $W_r^{(l)}$ is the weight for neighbor node $u$ connecting to node $v$ by an edge attributed with the relation $r \in R$, and $W_0^{(l)}$ is the weight for target node $v$. As shown above, the edge information is explicitly incorporated into a RGCN under the relation $r \in R$. The weight $W_r^{(l)}$ is a linear combination of the basis transformation. The initial node and edge representation can be found in Supplementary Table 5. The initial node (atom) and edge (bond) information used in RGCN.

By aggregating the information from each node through attention pooling, the embedding of a molecule can be obtained, and the molecular properties can be predicted by feeding the molecular embedding through three fully connected (FC) layers. The attention pooling is defined as follows:

$$\omega_v = \text{sigmoid}(W \cdot h_v + \text{bias}) \quad (2)$$

$$\text{molecular embedding} = \sum_{v=1}^{N} (\omega_v \cdot h_v) \quad (3)$$

where $W$ and bias are the trainable matrixes of the linear transformations in model training, $N$ is the number of nodes for a molecular graph, sigmoid() is an activation function that limits the attention weight $\omega_v$ of each node to between 0 and 1, $\omega_v$ is the attention weight of node $v$, and $h_v$ is the general feature of node $v$.

The key idea of perturbation-based methods is to generate masks to hide certain atoms/bonds/fragments from a GNN model in order to find which substructures may dramatically influence a model's prediction when they are absent in the input. However, the lack of chemical knowledge of the mask unit is prone to outputting confusing patterns of crucial molecular substructures that are difficult to be understood by chemists. Therefore, in this study, we adopt some common ways of splitting compounds in computational chemistry and used well-defined substructures as mask units.

As shown in Fig. 11d, BRICS, Murcko scaffolds and functional groups are used to generate different substructures to explain a molecular GNN model from different perspectives. As to BRICS, compounds are decomposed into the BRICS substructures based on a set of 16 bond types characterized with different chemical environments[25]. BRICS incorporates more elaborate medicinal chemistry concepts, and, for example, models explicit isosteric

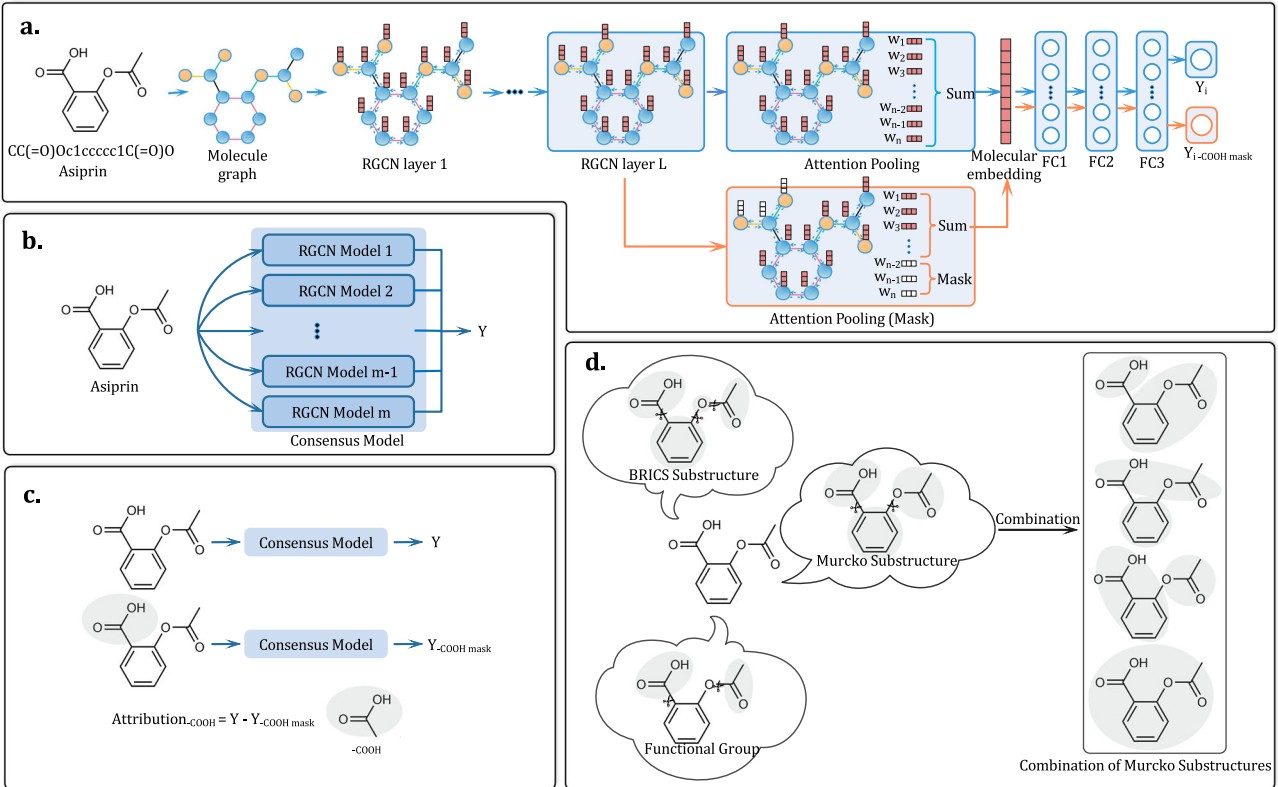

**Fig. 11 | The architecture of the consensus models and substructure mask explanation methods. a** The architecture of the RGCN sub-models; **b** The consensus model used for molecular property prediction; **c** The attribution calculation based on substructure mask explanation; **d** BRICS substructures, Murcko substructures, functional groups and the combination of Murcko substructures.

replacements for cyclic and acyclic cases. In addition, BRICS substructures can be assembled to form novel molecules by following a set of complementary rules. Molecular scaffold is another widely applied concept in medicinal chemistry and is mostly used to represent the core structures of bioactive compounds. A representative definition for scaffold is the Murcko framework proposed by Bemis and Murcko. Every molecule can be fragmented into a Murcko scaffold and the associated side chains[27]. Furthermore, the practice of analyzing molecules in terms of well-defined functional groups is also omnipresent in every branch of organic chemistry. In this study, we use 33 common functional groups collected in RDKit. Equipped with these three fragmentation methods, we should be able to take apart any organic molecule in terms of chemically meaningful substructures for any prediction task in medicinal chemistry and drug discovery.

Let us now elaborate on how the perturbation-based method works in detail. As mentioned earlier, the key idea is to identify a substructure (when missing from the input, e.g., see the masked carboxylic acid shown in Fig. 11a) that could dramatically influence a GNN's prediction. Since a GNN predicts molecular properties by feeding a molecular embedding to a properly trained FC layer, we need to give a masked molecule a proper embedding. To this end, we consider the following definition:

$$\text{molecular embedding}_{mask} = \sum_{v=1}^{N} (\omega_v \cdot h_v \cdot mask_v) \quad (4)$$

$$mask_v = \begin{cases} 1, & \text{if node v is not mask} \\ 0, & \text{if node v is mask} \end{cases} \quad (5)$$

where $N$ is the number of nodes, $\omega_v$ is the attention weight of node $v$, and $h_v$ is the general feature of node $v$.

In this study, we refer to the extent of the influence of a masked substructure on the overall prediction as the attribution. The attribution for each substructure may be determined as follows. Two GNN predictions before and after applying substructure masks onto a molecular graph are performed, and the difference between the predicted values is simply taken as the attribution:

$$Y = \sum_{i}^{m} Y_i \quad (6)$$

$$Y_{sub} = \sum_{i}^{m} Y_{i,sub} \quad (7)$$

$$\text{Attribution}_{sub} = Y - Y_{sub} \quad (8)$$

where $sub$ is the mask substructure, $m$ is the number of the RGCN models and 10 is used in this study, and $i$ is the $i^{th}$ RGCN model.

For a more intuitive visualization of interpretation analysis, we normalize the attribution scores of molecular substructures to Attribution_N with values ranging between 0 and 1.

After message passing, a node's hidden state gets updated by the information conveyed by adjacent atoms/bonds. Therefore, when we mask a node, it is not just about masking an atom but about masking the chemical environment centered on that atom. And the chemical environment contains information of adjacent atoms, while the information of the adjacent atoms weakens as the path to the central atom grows. Therefore, masking a node can be considered as masking the main information of the central atom, and also masking part of the information of the adjacent atoms/bonds. For ease of understanding,

in this article, the masking of a node is simply depicted as masking just the node itself.

## Combination of the BRICS and Murcko substructures

Chemical substructures often do not exhibit biological activity by themselves alone, while the manifestation of biological activity emerges when certain chemical substructures in a compound interact together. For example, a single amino group does not pose a risk of mutagenicity, but it becomes a genotoxic group with a high risk of mutagenicity when it is attached to a benzene ring. For the sac of ensuring calculation efficiency, SME does not adopt Shapley values to fairly assign attribute values to different substructures, but only calculates attribute values based on the changes between predicted values, which may overlook contribution by some substructures. In order to solve the above problems, we adopt a compensation strategy: find the most positive/negative substructures through the combination of substructures to avoid overlooking some important substructures, and thus explore the SAR more comprehensively. We randomly combined and masked the BRICS substructures and Murcko substructures of a molecule, and calculated the attribution for each substructure combination. Some representative combinations of the Murcko substructures of Aspirin are provided in Fig. 1d, and there are 7 combinations base on 3 substructures. As the number of fragments increases, the number of possible substructure combinations increases exponentially and leads to extremely large cost. Hence, we adopt a suboptimal solution in this study that a maximum of 100 combinations of BRICS substructures and 100 combinations of Murcko fragments are analyzed per molecule. The current combination method is a simple random combination, and advanced algorithms such as Monte Carlo tree search will be adopted to explore the attribution of different fragment combinations more efficiently in the future.

## Model construction and evaluation

Each dataset is randomly split into the training set, validation set, and test set by a ratio of 8:1:1. First, we conduct the hyperparameter optimization based on the predictive performance on the validation set. And the Tree Parzen Estimator (TPE) algorithm in Hyperopt (version 0.2.7) is used for the hyperparameter optimization in this study. The optimized hyperparameters are then used to build the RGCN submodels with different random seeds, and the ten sub-models are integrated to build the consensus model. The model construction is implemented using python (3.7) with dgl-cuda11.0(0.7.1), hyperopt (0.2.7) and pytorch (1.11.0). The data processing and metrics calculation are implemented using python (3.7) with scikit-learn (1.0.2), numpy (1.21.5) and pandas (1.3.5). The detailed information about the hyperparameters is listed in Supplementary Table 6. The regression tasks are evaluated by the squared determination coefficient ($R^2$), and the classification tasks are evaluated by the area under the receiver operating characteristic curve (ROC-AUC).

## Reporting summary

Further information on research design is available in the Nature Portfolio Reporting Summary linked to this article.

## Data availability

The ESOL, Mutagenicity, hERG and BBBP datasets used in this study and the data generated in this study are available at https://doi.org/10.5281/zenodo.7707093[66]. Supplementary Data, including Supplementary Data 1, Supplementary Data 2, and Supplementary Data 3, is provided alongside this paper.

## Code availability

The Supplementary Data and codes of SME are available at https://doi.org/10.5281/zenodo.7707093[66].

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

## Acknowledgements

This study was financially supported by National Key R&D Program of China (2021YFF1201400), National Natural Science Foundation of China (22220102001), and Natural Science Foundation of Zhejiang Province of China (LD22H300001).

## Author contributions

T.H., C.Y.H., D.C., and Z.W. designed the research study. Z.W. developed the method and wrote the code. Z.W., J.W., H.D., D.J., Y.K., D.L. P.P., and Y.D. performed the analysis. Z.W., T.H., C.Y.H., and D.C wrote the paper. All authors read and approved the manuscript.

## Competing interests

The authors declare no competing interests.
