## [Peer Review File · Nature Communications]

Chemistry-intuitive explanation of graph neural networks for molecular property prediction with substructure maskingREVIEWER COMMENTS

Reviewer #1 (Remarks to the Author):

This paper provides an explanation based XAI method which focuses on GNN models for molecular property prediction. The introduction briefly discusses current developments in the field and highlights the research problem well. The authors demonstrate advantages and limitations of their new SME approach using three examples; one regression task (solubility prediction) and two classification tasks (genotoxicity and cardiotoxicity prediction). A RGCN model architecture is used for the evaluation of all three tasks.

At a high-level, I find this to be a valuable contribution to this emerging area of XAI in chemistry. We note there is a similar manuscript to this one from May of this year. <https://chemrxiv.org/engage/chemrxiv/article-details/633731d1f764e6e535093041> proposing a similar substructure attribution method. And of course GNNExplainer is quite similar to the method here. The introduction points out that "actionable and insightful" explanations are important. I agree that actionable and insightful – meaning it is interpretable for a chemist – are both essential. On actionability, I believe the argument is that the structure optimization approach can be actionable. However, this kind of strays beyond an "explanation," since the authors make use of a slightly different task of doing molecular optimization. The actual explanation is a weight per substructure and the molecular optimization is an extra step. I think it's fine to do this in the same paper, but the authors should make sure to compare with other optimization methods out there – especially local ones like STONED or VAEs or GAs etc. If actionability is important to the authors – and I agree- then they should also discuss a bit of another popular actionable explanation method called counterfactuals (<https://pubs.rsc.org/en/content/articlehtml/2022/sc/d1sc05259d>) and maybe less on contrasting with other feature attribution methods. It is interesting that there seems to be some overlap between all these methods: doing substructure explanations and then optimization to find a similar but better molecule is quite like the counterfactual and contrastive explanations methods. I have some specific comments below:

The manuscript is relatively long - I wonder if it some content could be streamlined or moved to SI.

Introduction is neatly arranged to identify research gaps and the objectives of this paper. In the 3rd paragraph it is mentioned that, "...we argue that a molecular graph offers a more intuitive way for molecular representation...". However, whilst many examples show the usefulness of subgraph structures in molecular XAI it is not clear to the reviewer how this argument is supported in the paper. There are competing ideas about text, coordinates, SMILES, and it is not quite settled on which is best.

The authors highlight 5 existing XAI categories in the introduction section: gradient/feature based, decomposition, surrogate models, generation-based and perturbation based. However, no generation-based (GNN-explained?) and perturbation based are only lightly discussed with references to current work.

Table 1 lists the details of the three datasets used in training. However, it was not clear where exactly the data is from and if any pre-processing/splits were done.

The discussion on the RGCN model is well written. However, whilst this section is clear to a reader familiar with GNNs, it is not perfectly clear how a molecule is mapped onto a graph (e.g., how are bonds represented, how are atoms turned into nodes, and how is stereochemistry treated). The range of edges ($r \in R$) in equation 1 is unclear.

Hyperparameters of the RGCN model are unclear. For example, Figure 1A illustrates 3 fully connected layers, however, according to the text only one FC layer is used.

The section 2.2.4 refers to three substructure labels C31+ C32+ C33. But these labels are not shown in Figure 1D.

In section 3.1.1 it is mentioned that "... solubility is an intuitive concept taught in an introductory organic chemistry course ..." – surprisingly humans aren't that good at predicting solubility: <https://jcheminf.biomedcentral.com/articles/10.1186/s13321-017-0250-y>

One of the main critiques on the results sections is that, the applicability of SME is highly dependent on the performance of the trained model. As SME is suggested not only as an interpretation method but also as a structure optimization method, this can be a limitation. As shown in the results section, SME's substructure analysis aligns well with chemical intuition. However, it can be argued that how SME can be used in instances where prior chemical knowledge is lacking or the model performance is poor – how do we tell if/when it's working correctly

Typo in Figure 2 caption: "(A) The attribution visualization of compound 1, (B) The attribution visualization of compound 1 compound 2 "

Also it is recommended that molecular names are used instead of compound 1 and 2 (or something more memorable)

It is unclear which molecular substructure identification method from BRICS, Murcko and functional groups are useful in applications. How do we know which is correct and who is meant to consume this information? An organic chemist? An ML expert?

Structural optimization of molecules is another heavily studied field. References to existing work is important such that the readers are aware of the importance and challenges associated with structural optimization. The actual method to go from substructures to real molecules figure 3 is unclear to me – was it done by hand or was there a substructure library from which to select options?

Citation to the the list of Toxicophores and Detoxifying groups is not given from fig 4. Maybe figure 4 could be somehow compared more directly to the later results?

According to the results shown in figure 5 and as discussed by the authors the performance of SME in binary classification tasks seem to be poor. This raises the question if the SME is generalizable or not. Furthermore, the authors suggest the limitation of SME in such instances can be circumvented by considering individual fragments. Again this limits the applicability of the SME method. Therefore, the contribution of the method appear to be less impactful.

Authors mention that SME explanations combined with expert knowledge is needed in better structural optimization, specifically in identifying unreliable predictions. It is further mentioned that additional data can be used to improve the performance of SME. Therefore, it can be argued that the applicability of the SME method is highly limited when data are scarce, which is most chemistry related applications and the explanations generated from SME are not complete explanations although they are actionable.

Section 3.3.2 state that three ways of naturally reducing the hERG toxicity of molecules are,

1. adding a functional group with average attribution < 0
2. removing a functional group with an average attribution > 0
3. replacing a functional group with more negative attribution

The reviewer has the following questions based on the above statement.

1. How is an "average" attribution is quantified?
2. What is the threshold for selecting a "more negative" functional group based on the results?
3. Will simply removing an "average" negative group ensure reduction in toxicity?

Reviewer #2 (Remarks to the Author):

Reviewer has background in AI and machine learning research, focusing on learning algorithms that can capture and explain complex industrial processes, but reviewer also has publications co-written in the area of cheminformatics.

The paper presents an ensemble approach for better understanding molecule structure properties of solubility, mutagenicity and hERG toxicity. These three molecule properties are well understood, and are used in the paper to serve as a well-recognizable cases for chemists' view of understanding molecule structures that are drug candidates. The suggested approach matches these properties with entire (global) molecule structures based on molecule graphs, as compared to alternative approaches that matches only parts of molecules to explain molecule properties. The ensemble approach presented is based on three established building blocks of chemists' domain knowledge, including BRICKS substructures and Murcko scaffolds.

Not being a domain expert, it seems to me that the chosen baselines and related work is appropriate, the domain data and knowledge usage is very relevant. Most references to related work in the chosen problem area recent, as well as the connection to recent research in graph networks.

The paper is well-written and provides a high degree of completeness, both in positioning the content, connection with the domain, and the completeness and validity of the approach. Conclusions follow the content well.

Reviewer has a few remarks:

- At several instances the text motivates the need for matching global molecule properties, rather than using local features only. Consolidating this argumentation to one more powerful and well-placed instance would serve readability overall.
- Expand on the ensemble style of using BRICKS, Murcko etc. They are complementary, but how is consensus established when using them together. In which way do they complement each other to get a better result than otherwise/separate?
- The paper does not really address the limitations of the approach. The three well-chosen evaluation cases (solubility, mutagenicity and hERG toxicity) applies well, likely. How about generic chemists' understanding, also outside of these cases? To be able to state that explainability holds in general, to explain molecules in a chemist view and understanding, a much stronger validation would be needed, I think. Collecting data for validation about how well chemists are helped by the explanations for mining structure is a clear need, I think.
- How about sensitivity to the training data? There is some discussion on the consequence of imbalanced data, but very little.
- The possibility for replication of the work is well-served, with hyper-parameters clearly listed and with a Github repository. Still, the motivation for final hyper-parameter settings and choices are not well-explained.

Reviewer #3 (Remarks to the Author):

Comments to the author according to guideline for referees:

Key results

Your overview of the key messages of the study, in your own words, highlighting what you find significant or notable. Usually, this can be summarized in a short paragraph:

The presented approach derives chemically intuitive explanations for predictions of small molecules obtained from graph neural networks (and ensembles composed of such models). Deployed calculations facilitate a novel modification scheme of the attention layer, masking the contribution of meaningful substructures. Recorded difference between the prediction with the original and masked attention layer represent the feature importance. Utilized substructures can be freely chosen, but exemplary calculations for solubility, toxicity, and hERG activity were carried out using BRICS, Murcko scaffolds and the corresponding functional groups.

Validity

Your evaluation of the validity and robustness of the data interpretation and conclusions. If you feel there are flaws that prohibit the manuscript's publication, please describe them in detail:

Overall, conclusions drawn from the presented results are meaningful and correct, however I disagree that masking nodes reveals the importance of corresponding atoms, as after the message passing steps, nodes do not represent atoms, but molecular environments centered at the nodes. For example, masking a node originating from an amine group does not remove the information from a connected node, that it is adjacent to a nitrogen atom. Keeping this in mind, the explanations are still meaningful for the user, but it should be clarified in the text.

Significance

Your view on the potential significance of the conclusions for the field and related fields. If you think that other findings in the published literature compromise the manuscript's significance, please provide relevant references:

Presented work adds value to the understanding of molecular predictions, however due to the abundance of existing and established explanatory approaches, it will not revolutionize the field of XAI, but represent a useful tool for the cheminformatics community.

The strength of proposed method also poses its greatest limitation: As chemically intuitive explanations are enforced, patterns learned from data-artifacts, which are intrinsically not chemically meaningful, may remain concealed. Hence SME, which excels in communicating results with non-ML-experts, and structure agnostic models for the validation of models are complementary in nature.

Data and methodology

Your assessment of the validity of the approach, the quality of the data, and the quality of presentation. We ask reviewers to assess all data, including those provided as supplementary information. If any aspect of the data is outside the scope of your expertise, please note this in your report or in the comments to the editor. We may, on a case-by-case basis, ask reviewers to check code provided by the authors (see this Nature editorial for more information).

Reviewers have the right to view the data and code that underlie the work if it would help in the evaluation, even if these have not been provided with the submission (see this Nature editorial). If essential data are not available, please contact the editor to obtain them before submitting the report.

The overall methodology is valid, but as the authors already mentioned, explaining molecules split into multiple fragments may yield unexpected results. The rationale behind this issue is that calculating the importance as difference to the complete molecule violates criteria from the Shapley formalism, which ensure a fair attribution to each player (or feature). While the consequences are sufficiently discussed, it would be worth mentioning the origin of this issue.

Analytical approach

Your assessment of the strength of the analytical approach, including the validity and comprehensiveness of any statistical tests. If any aspect of the analytical approach is outside the scope of your expertise, please note this in your report or in the comments to the editor:

Statistical tests carried out in this study (correlation of average attribution and change of prediction) are meaningful and valid. However, the assessment of the explanation values should be carried out in a more systematic and objective manner (See suggested improvements).

Suggested improvements

Your suggestions for additional experiments or data that could help strengthen the work and make it suitable for publication in the journal. Suggestions should be limited to the present scope of the manuscript; that is, they should only include what can be reasonably addressed in a revision and exclude what would significantly change the scope of the work. The editor will assess all the suggestions received and provide additional guidance to the authors.

Major:

Comparing atom highlighting with substructure highlighting, where each atom is colored according to the total contribution of the substructure overstates the clear contrast between positive and negative contributions, as the absolute values are greater and therefore more dominant in their coloring. (Also, attributions from the atom mask are assumingly not derived according to the Shapley formalism, and hence are limited in their interpretability.) While this could be mentioned, a more subjective method to evaluate the explanatory power could be deployed, as presented by Rao et al. (<https://doi.org/10.48550/arXiv.2107.04119>). This would also address the more common use-case, where predictive performances of assessed models are mediocre.

Minor:

- Colored substructure highlighting should not have colored atom labels.
- Red/Green color-grading not accessible for visually impaired readers.
- Distributions of attributions should be represented in a more established depiction, such as a box- or violin-plot.
- Ln 450. [...] 0.003 to the amino group.
- Ln 492. You mean the mutagenicity dataset?
- Ln 625/646/653. Figure 12x?

Optional:

Substructure highlighting may benefit from using code presented at:
<http://rdkit.blogspot.com/2020/10/molecule-highlighting-and-r-group.html>

Clarity and context

Your view on the clarity and accessibility of the text, and whether the results have been provided with sufficient context and consideration of previous work. Note that we are not asking for you to comment on language issues such as spelling or grammatical mistakes.

The manuscript is well written and easy to access.

References

Your view on whether the manuscript references previous literature appropriately.

Ref 24: The only source I found (openreview.net) marked this work as rejected. It should be removed.

In the listing of explanatory approaches "Improving Molecular Graph Neural Network Explainability with Orthonormalization and Induced Sparsity" from Henderson et al. could be mentioned.

Reviewer 1:

Comments:

This paper provides an explanation based XAI method which focuses on GNN models for molecular property prediction. The introduction briefly discusses current developments in the field and highlights the research problem well. The authors demonstrate advantages and limitations of their new SME approach using three examples; one regression task (solubility prediction) and two classification tasks (genotoxicity and cardiotoxicity prediction). A RGCN model architecture is used for the evaluation of all three tasks.

At a high-level, I find this to be a valuable contribution to this emerging area of XAI in chemistry. We note there is a similar manuscript to this one from May of this year. <https://chemrxiv.org/engage/chemrxiv/article-details/633731d1f764e6e535093041> proposing a similar substructure attribution method. And of course GNNExplainer is quite similar to the method here. The introduction points out that “actionable and insightful” explanations are important. I agree that actionable and insightful – meaning it is interpretable for a chemist – are both essential. On actionability, I believe the argument is that the structure optimization approach can be actionable. However, this kind of strays beyond an “explanation,” since the authors make use of a slightly different task of doing molecular optimization. The actual explanation is a weight per substructure and the molecular optimization is an extra step. I think it’s fine to do this in the same paper, but the authors should make sure to compare with other optimization methods out there – especially local ones like STONED or VAEs or GAs etc. If actionability is important to the authors – and I agree- then they should also discuss a bit of another popular actionable explanation method called counterfactuals (<https://pubs.rsc.org/en/content/articlehtml/2022/sc/d1sc05259d>) and maybe less on contrasting with other feature attribution methods. It is interesting that there seems to be some overlap between all these methods: doing substructure explanations and then optimization to find a similar but better molecule is quite like the counterfactual and contrastive explanations methods.

Reply:

We thank the reviewer for encouraging and helpful comments. All the three recommended articles are examples of excellent work on XAI in cheminformatics. For simplicity, we shall refer to these two

additional works: (<https://pubs.rsc.org/en/content/articlehtml/2022/sc/d1sc05259d>) as STONED-Counterfactuals, and (<https://chemrxiv.org/engage/chemrxiv/article-details/633731d1f764e6e535093041>) as STONED-Lime in the rest of this response. Indeed, we already cited ‘GNNExplainer’ and STONED-Counterfactuals in our original manuscript and acknowledged it. We also thank the reviewer for pointing out the comparison to the other references based on STONED. We have now cited the new references (STONED-Lime) in the revised manuscript.

Note, STONED-Counterfactual belongs to the XAI category of ‘perturbation-based methods’ while STONED-Lime belong to the category of ‘surrogate methods’.

GNNExplainer can combine edges to form subgraph to obtain subgraph-level explanations in a post-processing manner. However, the important edges in their explanations are not guaranteed to be connected and hence the formed subgraph (for XAI purpose) is not guaranteed to be chemically meaningful. We added this description to the Introduction as follows:

“GNNExplainer and PGExplainer provide subgraph-level explanations by combining nodes or edges to form subgraphs through post-processing, but the important nodes or edges highlighted by these methods are not guaranteed to be connected as one fragment. SubgraphX can identify connected subgraphs with Monte Carlo tree search. However, due to the complexity of chemical rules, the fragments uncovered by these perturbative methods are often not chemically meaningful and prone to generating confusing or even frustrating interpretations for chemists.”

Next, we briefly compare STONED to the proposed SME. STONED is a novel method that can quickly generate a series of highly similar molecules with respect to a reference molecule. Hence, STONED-Counterfactuals and STONED-Limed both exploit STONED to enumerate many similar molecules and attempt to uncover ‘counter examples’ with respect to the given molecule of interest. Analyzing the SAR on these molecules allow chemists to arrive at an attribution of the key changed substructure in counter examples.

Conceptually, SME works differently. SME obtains the attribution value of a substructure by masking the substructure in the molecule, so the way SME obtains the attribution value does not need to generate a bunch of similar molecules for the molecule. However, we believe that both SME and STONED could be useful for the purpose of molecular optimizations. Hence, we thank the reviewer once again to bring up STONED which further inspires us that we can actually combine STONED and (SME attributed) functional groups to either generate or optimize molecules for specific properties. In the

revised manuscript, we added a description about STONED and how STONED-based interpretation methods can guide structure optimization as follows:

“Since the Superfast Traversal, Optimization, Novelty, Exploration and Discovery (STONED) method enables rapid exploration of chemical space without a pre-trained generative model, some interpretability methods combined with STONED can guide structure optimization.”

Comment 1: *The manuscript is relatively long - I wonder if it some content could be streamlined or moved to SI.*

Reply:

We thank the reviewer for the advice. We have re-organized the content as suggested.

Comment 2: *Introduction is neatly arranged to identify research gaps and the objectives of this paper. In the 3rd paragraph it is mentioned that, “...we argue that a molecular graph offers a more intuitive way for molecular representation...”. However, whilst many examples show the usefulness of subgraph structures in molecular XAI it is not clear to the reviewer how this argument is supported in the paper. There are competing ideas about text, coordinates, SMILES, and it is not quite settled on which is best.*

Reply:

The rationale behind our argument is that many chemists tend to reason about molecular and chemical properties in terms of simple molecular graphs and associated substructures as an intuitive crutch. Nevertheless, we agree that other competing molecular representation, such as SMILES, could serve exactly the same purpose very well; although we do believe most chemists (especially experimentalists) should be more comfortable with 2D molecular graphs but, obviously, we do not have hard evidence. For non-deep-learning methods, we often rely on handcrafted physiochemical descriptors as molecular features. For these cases (which we implicitly neglected in our original manuscript), it is also possible that other attribution analyses may derive an intuitive explanation that is complementary and equally competitive to that offered by molecular graph approach, a more popular featurization in the era of deep learning. In light of this thoughtful comment by the reviewer, we decide to modify the message as follows (in the revised manuscript):

“A common choice is molecular graphs, which intuitively correspond to chemical structures (nodes correspond to atoms, edges correspond to chemical bonds, and subgraphs correspond to substructures).

This observation indicates molecular graph as a suitable representation with great potential for clear interpretability.”

Comment 3: *The authors highlight 5 existing XAI categories in the introduction section: gradient/feature based, decomposition, surrogate models, generation-based and perturbation based. However, no generation-based (GNN-explained?) and perturbation based are only lightly discussed with references to current work.*

Reply:

Compared with other methods, generation-based approach is still less explored. To the best of our knowledge, XGNN was the only existing generation-based method. Hence, we added this as follows:

“To the best of our knowledge, XGNN is the only generation-based method that renders high-level explanations of GNNs by generating graph patterns to maximize a certain prediction.”

We point out the main idea of the permutation method: identify key features by monitoring the degrees of changes in the predictions through perturbing different input features. Due to the length restriction on the article, we only list some representative works here (GNNExplainer, PGExplainer and SubgraphX) and discuss the work, SubgraphX, most related to our work in detail. And we added a comment as follows:

“As summarized above, all existing methods could potentially be improved to better suit the needs of chemistry community, and a more detailed account on all these methods can be found in reference 14.”

Comment 4: *Table 1 lists the details of the three datasets used in training. However, it was not clear where exactly the data is from and if any pre-processing/splits were done.*

Reply:

Detail on the datasets can be found in Table S1. And we have stated this in the paper: *“The detail on these 4 datasets is summarized in Table 1 and further detail on the data curation can be found in Table S1.”* Every dataset is randomly split into the training set, validation set, and test set by a ratio of 8:1:1. And we have stated this in 2.3 Model Construction and Evaluation: *“Each dataset is randomly split into the training set, validation set, and test set by a ratio of 8:1:1.”*

Table S1 is shown below, and the source literature of the dataset is also added.

Table S1. The detailed information of different datasets.

Category	Description	The number of molecules
ESOL	Small dataset consisting of water solubility data for 1111 compounds ¹ . The duplicated molecules and the molecules with conflicting label values are excluded.	1111
Mutagenicity	The training set for model building was collected from four papers. The data set for external validation was extracted from the Web site of Lazar toxicity predictions. The entire database was prepared as following. First, apart from the four false SMILES strings, duplicate molecules were removed from the five sources by using canonical SMILES. Second, molecules without clear E or Z configuration were removed. Third, inorganic compounds were omitted from the data set. The last step was to eliminate the tautomers and compounds with molecular weight less than 40 or more than 800 in the data set. When doing the data set curation, we followed one principle. For a given compound, if the experimental mutagenicity data varied in different sources, the compound was cleared out. For compounds without defined steric configuration or tautomers, if the experimental mutagenicity data was alike, then only one structure was kept, and the others were deleted. ²	7672
hERG	The original chemicals with experimental IC ₅₀ values are collected from a publication ³ and ChEMBL database. Molecules with IC ₅₀ ≤ 10 μM are classified as hERG blockers, and molecules with IC ₅₀ > 10 μM are classified as hERG nonblockers. Inorganic compounds, noncovalent complexes and mixtures are removed from the data set. The duplicated molecules and the molecules with conflicting label values were excluded.	9876

	The dataset is from ADMET lab 2.0. ⁴	
BBBP	Category 0: BBB-; Category 1: BBB+;	1859
	The molecules were divided into BBB+ and BBB-classes with	
	$\log_{BB} \geq -1$ and $\log_{BB} < -1$, respectively.	

Comment 5: *The discussion on the RGCN model is well written. However, whilst this section is clear to a reader familiar with GNNs, it is not perfectly clear how a molecule is mapped onto a graph (e.g., how are bonds represented, how are atoms turned into nodes, and how is stereochemistry treated). The range of edges ($r \in R$) in equation 1 is unclear.*

Reply:

We thank reviewer for the detailed comment. We have now added information on initial node and edge (atom and bond) representation in Table S2. R here represents the set of edge types, and we added the corresponding description as follows:

“R denotes the set of edge types”

“The initial node and edge representation can be found in Table S2”

Table S2. The initial node (atom) and edge (bond) information used in RGCN.

Node feature	Size	Description
Atom symbol	16	[B, C, N, O, F, Si, P, S, Cl, As, Se, Br, Te, I, At, metal] (one-hot)
degree	6	number of covalent bonds [0,1,2,3,4,5] (one-hot)
formal charge	1	electrical charge (integer)
hybridization	6	[sp, sp ² , sp ³ , sp ^{3d} , sp ^{3d2} , other] (one-hot)
aromaticity	1	whether the atom is part of an aromatic system [0/1] (one-hot)
hydrogens	5	number of connected hydrogens [0,1,2,3,4] (one-hot)
chirality	1	whether the atom is chiral center [0/1] (one-hot)
chirality type	2	[R, S] (one-hot)
Edge feature	Size	Description
bond type	4	[single, double, triple, aromatic]
conjugation	1	whether the bond is conjugated [0/1]
ring	1	whether the bond is in ring [0/1]

Comment 6: *Hyperparameters of the RGCN model are unclear. For example, Figure 1A illustrates 3 fully connected layers, however, according to the text only one FC layer is used.*

Reply:

We thank the reviewer pointing out this point, and the text is now corrected as follows:

“By aggregating the information from each node through attention pooling, the embedding of a molecule can be obtained, and the molecular properties can be predicted by feeding the molecular embedding through three fully connected (FC) layers.”

In addition, we have added more detailed information about hyperparameters of RGCN in Table S3.

And we added a description as follows:

“And the Tree Parzen Estimator (TPE) algorithm in Hyperopt (version 0.2.7) is used for the hyperparameter optimization in this study.”

Table S3 is as follows:

Table S3. The hyperparameters of different models.

Model	Parameters to be optimized	Package
ESOL	the number of nodes of each RGCN hidden layer: [64, 128, 256]	DGL 0.7.1
	the number of RGCN hidden layer: [2, 3]	
	the number of nodes of each FC hidden layer: [64 , 128, 256]	
	the dropout rate of each RGCN hidden layer: [0, 0.1, 0.2, 0.3, 0.4, 0.5]	
	the dropout rate of each FC hidden layer: [0, 0.1 , 0.2, 0.3, 0.4, 0.5]	
	the learning rate: [0.003 , 0.001, 0.0003, 0.0001]	
	the number of epochs: 500	
	the patience of early stop: 30	
Mutagenicity	the number of nodes of each RGCN hidden layer: [64, 128, 256]	DGL 0.7.1
	the number of RGCN hidden layer: [2, 3]	
	the number of nodes of each FC hidden layer: [64, 128 , 256]	
	the dropout rate of each RGCN hidden layer: [0, 0.1, 0.2, 0.3, 0.4 , 0.5]	
	the dropout rate of each FC hidden layer: [0 , 0.1, 0.2, 0.3, 0.4, 0.5]	

	the learning rate: [0.003, 0.001 , 0.0003, 0.0001]	
	the number of epochs: 500	
	the patience of early stop: 30	
	the number of nodes of each RGCN hidden layer: [64 , 128, 256]	
	the number of RGCN hidden layer: [2, 3]	
	the number of nodes of each FC hidden layer: [64, 128 , 256]	
hERG	the dropout rate of each RGCN hidden layer: [0, 0.1, 0.2 , 0.3, 0.4, 0.5]	DGL 0.7.1
	the dropout rate of each FC hidden layer: [0, 0.1 , 0.2, 0.3, 0.4, 0.5]	
	the learning rate: [0.003, 0.001, 0.0003 , 0.0001]	
	the number of epochs: 500	
	the patience of early stop: 30	
	the number of nodes of each RGCN hidden layer: [64, 128, 256]	
	the number of RGCN hidden layer: [2 , 3]	
	the number of nodes of each FC hidden layer: [64, 128 , 256]	
BBBP	the dropout rate of each RGCN hidden layer: [0, 0.1, 0.2 , 0.3, 0.4, 0.5]	DGL 0.7.1
	the dropout rate of each FC hidden layer: [0, 0.1, 0.2, 0.3, 0.4 , 0.5]	
	the learning rate: [0.003, 0.001 , 0.0003, 0.0001]	
	the number of epochs: 500	
	the patience of early stop: 30	

Bold hyperparameters represent optimized hyperparameters.

Comment 7: *The section 2.2.4 refers to three substructure labels C31+ C32+ C33. But these labels are not shown in Figure 1D.*

Reply:

We thank review for pointing this out. C31+ C32+ C33 are actually not substructure labels, and they refer to the conventional mathematic notation C_k^n (of choosing k combinations out of n choices). In this case, we look at all combinations of choosing from 3 Murcko substructures. For example, C32 means that there are 3 combinations where two of the three fragments are randomly selected for combination, and $C32 = 3$. We apologize that we simplify the notation here, and cause confusions. Since the number

of combinations has been shown in Figure 1D, in order to avoid misunderstandings, we now delete the unconventional notation C31+ C32+ C33 from the text as follows:

“Some representative combinations of the Murcko substructures of Aspirin are provided in Figure 1D, and there are 7 combinations based on 3 substructures.”

Comment 8: *In section 3.1.1 it is mentioned that “... solubility is an intuitive concept taught in an introductory organic chemistry course ...” – surprisingly humans aren’t that good at predicting solubility: <https://jcheminf.biomedcentral.com/articles/10.1186/s13321-017-0250-y>*

Reply:

We fully agree with the reviewer that predicting solubility remains a challenge, but chemists have explored and broadly understood the effect of different functional groups on water solubility (although the substructure-level understanding may not always translate into a successful global prediction.) To be more precise about what we imply, let us give an example. For most cases, the hydroxyl group generally makes a molecule more soluble than the methyl group. Nevertheless, we agree with the review that we should be more cautious with our wordings. We have modified the description and cited the above work as follows:

“Although accurate prediction of water solubility remains a difficult problem, chemists have explored and broadly understood the effect of a large number of functional groups on water solubility. Hence, solubility is often used to validate model explanation methods and it provides a good setting for testing SME.”

Comment 9: *The Typo in Figure 2 caption: “(A) The attribution visualization of compound 1, (B) The attribution visualization of compound 1 compound 2”*

Reply:

We thank the reviewer, and have corrected the typo.

Comment 10: *Also it is recommended that molecular names are used instead of compound 1 and 2 (or something more memorable)*

Reply:

We thank the review for a great advice, and we have added canonical SMILES for molecules in Table 2, where all compounds appearing in the main text are carefully numbered. Considering that readers can easily identify the corresponding molecule by looking up Table 2, we still use “compound + number” to denote the compound. The description and Table 2 are added in section 2.1 Datasets as follows:

“In addition, the canonical SMILES of the compounds for analysis are shown in Table 2.”

Table 2. The canonical SMILES of the compounds.

Molecule	Canonical SMILES
Compound 1	<chem>CC1CCC(C(C)C)C(O)C1</chem>
Compound 2	<chem>COc1ccc(C(O)(c2cncnc2)C2CC2)cc1</chem>
Compound 3	<chem>Nc1c(C(=O)O)cc([N+](=O)[O-])c2c1C(=O)c1ccccc1C2=O</chem>
Compound 4	<chem>Cc1ccc(N)cc1[N+](=O)[O-]</chem>
Compound 5	<chem>COc1cc([N+](=O)[O-])ccc1N</chem>
Compound 6	<chem>[N-]=[N+]=Nc1ccc(F)c([N+](=O)[O-])c1</chem>
Compound 7	<chem>[N-]=[N+]=Nc1ccc(F)c([N+](=O)[O-])c1</chem>
Compound 8	<chem>O=[N+](O)c1ccc2c3ccccc3c3cccc4ccc1c2c43</chem>
Compound 9	<chem>NCc1ccc(F)c(C2CCN(C(=O)c3cccc(-c4nc(-c5cccs5)no4)c3)CC2)c1</chem>
Compound 10	<chem>NCc1ccc(F)c(C2CCN(C(=O)c3cc(C(=O)O)cc(-c4nc(-c5cccs5)no4)c3)CC2)c1</chem>
Compound 11	<chem>NCc1ccc(F)c(C2CCN(C(=O)c3cc(C(N)=O)cc(-c4nc(-c5cccs5)no4)c3)CC2)c1</chem>
Compound 12	<chem>COc1ccc(CCN2CCC(CCc3ccccc3OCCF)CC2)cc1</chem>
Compound 13	<chem>FCCOc1ccccc1CCC1CCN(CCc2ccccc2)CC1</chem>
Compound 14	<chem>FCCOc1ccccc1CCN1CCN(CCc2ccccc2Cl)CC1</chem>
Compound 15	<chem>FCCOc1ccccc1CCN1CCN(CCc2ccccc2)CC1</chem>
Compound 16	<chem>CCn1nc(Cc2ccc(C#N)cc2)cc1C1CCN(C[C@H]2CN([C@@H](C(=O)O)C(C)(C)C)C[C@@H]2c2cccc(F)c2)CC1</chem>
Compound 17	<chem>CCn1nc(Cc2ccc(S(C)(=O)=O)cc2)cc1C1CCN(C[C@H]2CN([C@@H](C(=O)O)C(C)(C)C)C[C@@H]2c2cccc(F)c2)CC1</chem>
Compound 18	<chem>Nc1ccc2nc(Cc3ccc(Oc4ccccc4)cc3)[nH]c2c1</chem>
Compound 19	<chem>CC(=O)Nc1ccc2nc(Cc3ccc(Oc4ccccc4)cc3)[nH]c2c1</chem>

Comment 11: *It is unclear which molecular substructure identification method from BRICS, Murcko and functional groups are useful in applications. How do we know which is correct and who is meant to consume this information? An organic chemist? An ML expert?*

Reply:

We appreciate this question, which helps us to shape a more comprehensive introduction of SME to potential users. In short, our response goes as follows.

Different schemes of fragmenting substructures may be suitable for different application scenarios. Users (we envision many of them would be chemists with some experiences using ML for their investigations) can make their own decisions about which method is more suitable in their case. In order to give less experienced users some suggestions, we now offer a more detailed description on various scenarios in which we believe a particular approach of fragmentation with SME can offer the most valuable insights in the revised manuscript. The details are as follows:

“Under each scenario, we recommend different fragmentation schemes to go along with SME for analysis. A succinct overview is provided below.

Scenarios 1. Mine the SAR for a specific molecule (local explanation): analyze the attributions of different substructures in a molecule, it would be good to consider all fragmentation schemes (i.e., a combination of BRICS substructures, Murcko substructures and functional groups);

Scenarios 2. Identify the most positive/negative components of a specific molecule: obtain the combined fragments with the most positive/negative attribution through the combination of BRICS substructures and Murcko substructures;

Scenarios 3. Mine the SAR for the desired properties (global explanation) on a statistical basis: analyze the functional group attributions on the whole dataset;

Scenarios 4. Provide guidance for structural optimization: compare the average attributions of different functional groups.

Scenarios 5. Molecule generation for desired properties: recombination of BRICS substructures with SME attribution scores.”

Comment 12: *Structural optimization of molecules is another heavily studied field. References to existing work is important such that the readers are aware of the importance and challenges associated*

with structural optimization. The actual method to go from substructures to real molecules figure 3 is unclear to me – was it done by hand or was there a substructure library from which to select options?

Reply:

We thank the reviewer for this useful question and we have clarified this part in the revised manuscript. In principle, there is a substructure library comprising all the fragmentations that are derived from the molecules in the training set with the appropriate attribution scores assigned by SME. Therefore, the structural optimization can be performed via replacing suitable fragments. In Fig. 3, though, we just change the methyl group to one of the functional groups in Figure 2C by hand as an illustration. We now add details to clarify potential misunderstandings like this, and also cited the STONED-based method that could combine with SME for structural optimization. Revised excerpts from the manuscript are as follows:

“Since the Superfast Traversal, Optimization, Novelty, Exploration and Discovery (STONED) method enables rapid exploration of chemical space without a pre-trained generative model, some interpretability methods combined with STONED can guide structure optimization.”

“To structurally optimize compound 1 to change its water solubility, we can manually change the methyl group to a functional group in Figure 2C.”

Comment 13: *Citation to the list of Toxicophores and Detoxifying groups is not given from fig 4. Maybe figure 4 could be somehow compared more directly to the later results?*

Reply:

We thank the reviewer for this useful reminder, and we have now cited the reference for the list Toxicophores and Detoxifying groups. Moreover, we have also taken the advices seriously and made the suggested comparisons to the later results in the revised manuscript. We added the corresponding toxicophores and detoxifying in Figure 5 and Figure 6 as follows:

Figure 5. The attribution visualization of compound 3.

Figure 6. The attribution visualization and structural optimization of compounds 4, 5 and 6.

Comment 14: According to the results shown in figure 5 and as discussed by the authors the performance of SME in binary classification tasks seem to be poor. This raises the question if the SME is generalizable or not. Furthermore, the authors suggest the limitation of SME in such instances can be circumvented by considering individual fragments. Again this limits the applicability of the SME method. Therefore, the contribution of the method appears to be less impactful.

Reply:

We appreciate this question, and we understand the reviewer's concern. In fact, we actually investigated and briefly discussed it in the original manuscript. As demonstrated in 3.2.2, SME can be easily extended

to estimate not only the attribution scores for individual fragments but also the scores for the combination of substructures. And as shown in Figure 6, though SME may underestimate the attributions of toxicophores in molecules when there are multiple (more than 1) toxicophores in the molecules. However, through analyzing the interplay of different substructures in a molecule, SME can still automatically mine and identify all essential toxicophores as shown in Figure 6 (compounds 4, 5 and 6). Although, to do this consistently for larger molecules in a high-throughput manner, more advanced searching algorithm should be used in conjunction with SME. Moreover, the functional groups attribution analysis demonstrated that SME can indeed extract a reasonable structure-activity relationship from the model. For example, as to Mutagenicity, the functional groups with the positive attributions are often the components of the identified toxic groups (-N=O, -NO₂ and -N), while those with the negative attributions (-SO₃H, and-CF₃) are identified to be detoxifying groups. The above results demonstrated that SME can handle binary classification problems with some calculation overhead. Moreover, the case study of hERG is also a classification problem in this study, and SME has also shown good performance in uncovering optimization strategies. To validate that SME can handle a variety of classification problems, we conducted one more experiment (in this revision) by applying SME to analyze and explain the GNN prediction of another property of great interest to medicinal chemists, the blood-brain barrier penetration (also a binary classification task). In our revised manuscript, we show that SME can also extract reasonable structure-activity relationship for BBBP, which further demonstrated that SME can generally handle classification problems with one additional example sufficiently distinct from the cases of Mutagenicity and hERG in our original study.

Comment 15: *Authors mention that SME explanations combined with expert knowledge is needed in better structural optimization, specifically in identifying unreliable predictions. It is further mentioned that additional data can be used to improve the performance of SME. Therefore, it can be argued that the applicability of the SME method is highly limited when data are scarce, which is most chemistry related applications and the explanations generated from SME are not complete explanations although they are actionable.*

Reply:

There are two parts in this question. First, is SME applicable when data are scarce? Since SME (and other explanation methods) are meant to explain what a GNN model learns from data. Therefore, the

sensibility of any explanation method critically depends on the reliability of the GNN model to be explained. Unfortunately, it is challenging for ML models to learn the true or complete causality when data are scarce. This challenge renders a model either making many unreliable predictions or only trustworthy in a limited context (i.e. not generalizable as it has not been trained on a representative set of molecules). This is a difficulty beyond the scope of this work. However, we stress again that this is a more fundamental limitation on machine learning itself rather than a metrics one should use to compare various explanation methods (SME or other competing approaches). Furthermore, we apologize that our remark “additional data can be used to improve the performance of SME” led to the misconception that SME is more data hungry than other alternative explanation methods. Rather, our intention was to say that all explanation methods could benefit from the availability of more data (which cover all the essential SARs in the training set and allows us to train a high-quality GNN model that any explanation method will have an easier time to yield sensible interpretations).

The second question regards whether SME (or, for the sake of this discussion, any explanation methods) would be useful to study chemistry related applications in which data scarcity is a common problem. Our response is that if one chooses to rely on a ML model to make predictions in this difficult scenario, then SME definitely helps experts to more quickly spot the unreliable parts of the ML model by analyzing the extracted SARs. For example, one can see conflicting assignments of attribution scores to the same fragments and a more detailed understanding on which of the extracted SARs (or attributed fragments) could be unreliable. When expert decides to perform structural optimizations then they can judiciously ignore certain SARs or fragment replacements to ensure they avoid unreliable information / predictions from the underlying GNN models.

In short, we summarize by stating that data scarcity poses a challenge for all explanation methods as this is a limitation on the underlying ML models to be explained. However, SME is not more data hungry than other competing approaches, and provides a transparent inspection that allows us to identify the source of unreliability of ML models in some cases (requiring expert’s own judgements at this point). Finally, we remind that we have tested SME on ESOL (1111), BBBP (1859), hERG (9876) and Mutagenicity datasets (7672) datasets and achieved good performance (ESOL with $R^2=0.927$, Mutagenicity with ROC-AUC=0.901, hERG with ROC-AUC=0.862, BBBP with ROC-AUC=0.919), which demonstrated the generalization of the SME.

Comment 16:

Section 3.3.2 state that three ways of naturally reducing the hERG toxicity of molecules are,

1. adding a functional group with average attribution < 0
2. removing a functional group with an average attribution > 0
3. replacing a functional group with more negative attribution

The reviewer has the following questions based on the above statement.

1. How is an “average” attribution is quantified?
2. What is the threshold for selecting a “more negative” functional group based on the results?
3. Will simply removing an “average” negative group ensure reduction in toxicity?

Reply:

We thank reviewer for these thoughtful questions. After reading the reviewer’s question, we realized that there was some ambiguity in the way we wrote the message. Our original intention is that SME can naturally provide three structural optimization strategies, but it does not necessarily imply an 100% certainty that every adopted optimization (based on SME) would surely reduce the toxicity of hERG. However, we hope the reviewer may agree that there is no 100% guarantee with anything derived from ML (especially for chemistry and related fields) most of the time if not all the time.

Therefore, we change the original expression *"the hERG toxicity of a compound can be naturally reduced in the following three ways"* to *"SME can naturally provide three structural optimization strategies to reduce hERR toxicity as follows"*

1. The “average” attribution is calculated based on the whole datasets. First, we calculated the attribution of functional groups in different molecules in the whole dataset. And the average attribution is then determined based on the attribution of functional groups in the whole dataset. For example, if the functional group appears 100 times in the molecules of the training set, then there will be 100 attributions, and the mean of the 100 attributions is the “average” attribution.
2. A “more negative” functional group is a group with a more negative average attribution. Since the average attribution is based on all data in the entire training set, molecules with an average score of 0.1 will be less likely to exhibit hERG toxicity in most cases than molecules with an average score of 0.3. For a substructure to be optimized, we may have multiple substructures that are more negative than it, and we should first try the more negative substructures among these negative substructures.

Therefore, we first use the average attribution as the threshold, and then we can preferentially select more negative fragments among the compatible fragments for structural optimization.

3. As mentioned above, the strategies mentioned are only guidelines for structural optimization based on the performance of the model on the entire dataset, and may not be completely accurate or optimal.

Reviewer 2:

Comments:

Reviewer has background in AI and machine learning research, focusing on learning algorithms that can capture and explain complex industrial processes, but reviewer also has publications co-written in the area of cheminformatics.

The paper presents an ensemble approach for better understanding molecule structure properties of solubility, mutagenicity and hERG toxicity. These three molecule properties are well understood, and are used in the paper to serve as a well-recognizable cases for chemists' view of understanding molecule structures that are drug candidates. The suggested approach matches these properties with entire (global) molecule structures based on molecule graphs, as compared to alternative approaches that matches only parts of molecules to explain molecule properties. The ensemble approach presented is based on three established building blocks of chemists' domain knowledge, including BRICKS substructures and Murcko scaffolds.

Not being a domain expert, it seems to me that the chosen baselines and related work is appropriate, the domain data and knowledge usage is very relevant. Most references to related work in the chosen problem area recent, as well as the connection to recent research in graph networks.

The paper is well-written and provides a high degree of completeness, both in positioning the content, connection with the domain, and the completeness and validity of the approach. Conclusions follow the content well.

Reply:

We gratefully appreciate your encouraging and helpful comments.

Comment 1: At several instances the text motivates the need for matching global molecule properties, rather than using local features only. Consolidating this argumentation to one more powerful and well-

placed instance would serve readability overall.

Reply:

We thank reviewer for the thoughtful suggestion, and we now provide a clear global use suggestion for SME. Moreover, we now summarize the scenarios that SME can be applied and offer some suggestions on which fragmentation strategy should be used in each case. We believe these suggestions will improve the overall readability and understanding the impact of SME. Our experimental design is based on the following scenarios, and the results illustrate the practicability of SME in these application scenarios. The section s3.1.1 and 3.2.1 show the application examples of SME in scenarios 1 and 2. The sections 3.1.2, 3.2.2&3.2.3 and 3.3 show the application examples of SME in scenarios 3 and 4. And the section 3.4 shows the application examples of SME in scenarios 5.

The suggestions are as follows:

“Under each scenario, we recommend different fragmentation scheme to go along with SME for analysis. A succinct overview is provided below.

***Scenarios 1.** Mine the SAR for a specific molecule (local explanation): analyze the attributions of different substructures in a molecule, it would be good to consider all fragmentation schemes (i.e., a combination of BRICS substructures, Murcko substructures and functional groups);*

***Scenarios 2.** Identify the most positive/negative components of a specific molecule: obtain the combined fragments with the most positive/negative attribution through the combination of BRICS substructures and Murcko substructures;*

***Scenarios 3.** Mine the SAR for the desired properties (global explanation) on a statistical basis: analyze the functional group attributions on the whole dataset;*

***Scenarios 4.** Provide guidance for structural optimization: compare the average attributions of different functional groups.*

***Scenarios 5.** Molecule generation for desired properties: recombination of BRICS substructures with SME attribution scores.*

”

Comment 2: - Expand on the ensemble style of using BRICKS, Murcko etc. They are complementary, but how is consensus established when using them together. In which way do they complement each other to get a better result than otherwise/separate?

Reply:

We thank reviewer for raising this point. BRICS and Murcko are different schemes that can split molecules into chemically meaningful molecular substructures, and SME consolidates different fragments by comparing the attribution scores, and an immediate benefit of this consolidation is that SME may determine the most positive/negative and chemically meaningful components in molecules. Let us give an example to illustrate this point. As shown in Figure 2.1 below, the most hydrophobic component in the molecule comes from Murcko fragments but the most hydrophilic component comes from BRICS fragments. Therefore, Murcko and BRICS can complement each other, enabling chemists to quickly focus on the key fragments with well-defined chemical implications.

Figure 2.1

Comment 3: *The paper does not really address the limitations of the approach. The three well-chosen evaluation cases (solubility, mutagenicity and hERG toxicity) applies well, likely. How about generic chemists' understanding, also outside of these cases? To be able to state that explainability holds in general, to explain molecules in a chemist view and understanding, a much stronger validation would be needed, I think. Collecting data for validation about how well chemists are helped by the explanations for mining structure is a clear need, I think.*

Reply:

We thank reviewer for pointing out this issue. Despite possessing multiple advantages and potential towards mining SAR information, structural optimization and de novo design, SME also has limitations like all previous methods. First, since SME (and other GNN-based interpretation methods) aims to explain what the GNN model has learned from the data, it's difficult for SME to mine reliable SAR

information when the GNN model has not learned the true or complete causality due to data paucity or data bias. Second, as chemically intuitive explanations are enforced by design, some subtle errors learned from data artifacts may remain concealed. Lastly, the current version of SME only supports the three substructures of BRICS, Murcko and functional groups, and does not allow the assessment of some other substructures such as bioisosteres etc.

In short, these limitations are either fundamental to all methods or a tradeoff between a structure agnostic approach and chemical-structure-dependent explanation method like SME as also briefly commented by Reviewer #3. Nevertheless, we stress that SME brings a fresh and complementary view on the interpretation methods (currently, dominated by structure agnostic approaches) on graph neural networks in the chemistry community. We advocate the benefits of SME outweigh its limitations as illustrated by many examples in our manuscript. For example, we show that SME could be applied to 4 key druggability properties tasks (ESOL, Mutagenicity, hERG and BBBP), which demonstrates that SME is a tool with great potential to assist medicinal chemists to mine SAR information for structure optimization and de novo design. We have added the above limitations in the **Conclusions** section as follows:

“Just like all other explanation methods for deep learning methods, SME also possesses several limitations. First, since SME (and other GNN-based interpretation methods) aims to explain what the GNN model has learned from the data, it is difficult for SME to mine reasonable SAR information when the GNN model has not learned the true or complete causality due to data paucity or data bias. Second, as chemically intuitive explanations are enforced, intrinsically chemically meaningless patterns learned from data artifacts may remain concealed. Lastly, the current version of SME only supports the three substructures of BRICS, Murcko and functional groups, and does not allow the assessment of some other substructures such as bioisosteres. Irrespective of these limitations (some can be easily overcome such as the last point mentioned above), SME provides intuitive and insightful interpretability for medicinal chemists based on chemically meaningful substructures. As most existing attribution methods for molecular GNNs in chemistry are directly borrowed from other fields and lacks the kind of chemical intuition that SME insists by design, we believe its benefits outweigh its drawbacks. Particularly, we demonstrate how SME help chemists in 4 key druggability properties tasks (ESOL, Mutagenicity, hERG and BBBP). In sum, SME is a tool with great potential to assist chemists to gain an appropriate understanding of why a GNN model makes certain prediction, and mine SAR information for structure

optimization and de novo design.”

“How about generic chemists’ understanding, also outside of these cases? To be able to state that explainability holds in general, to explain molecules in a chemist view and understanding, a much stronger validation would be needed, I think. Collecting data for validation about how well chemists are helped by the explanations for mining structure is a clear need, I think.”

It is difficult to make a very definite statement about how chemists can be helped by SME or other explanation methods for GNNs in a broad sense. Since there is no definite ‘metrics’ or ‘common benchmarks’ for evaluating interpreting methods for GNN property prediction models, we think collecting data for validation is difficult. Nevertheless, we believe our illustrative examples already cover many realistic scenarios in typical drug discovery projects. But we acknowledge that there are certainly more scenarios than what we had outlined in the previous manuscript. Hence, to more thoroughly address this comment, we decided to conduct one more experiment which illustrates how SME could help chemists in a way that is different from what we had done with all examples in the previous manuscript and we hope this could really illustrate the versatility of SME.

In this new example, we discuss how to use the structure-activity relationship derived from SME to develop further interpretation beyond SARs. By analyzing how models predict blood–brain barrier permeation (BBBP) task as an extra example, we show how chemists can mine SAR information through SME and use this valuable information to further expose favorable physiochemical properties these molecules must possess. Converting information on favorable molecular substructures to favorable physiochemical properties is an abstraction that may actually inspire further understanding of complex phenomena such as BBBP. We hope this example gives a new perspective on SME’s versatility. The details of this new experiment are further described below.

Figure 3.1

Through the analysis of functional group attribution generated by SME, we can construct a desired relationship between molecular functional groups and the BBBP as shown in above **Figure SME 3.1**. Similarly, by constructing different prediction models for characteristic physicochemical properties (such as LogP and TPSA in the figure), we can construct another desired relationship linking molecular functional groups and physicochemical properties. Subsequently, we then establish the relationship between BBBP and the basic physicochemical properties, providing a deeper understanding on what molecular properties (substructure or its effects on physiochemical attributes) hold the key to having the desired properties.

The section about BBBP is as follows:

“3.5 How can SME establish SAR information with deeper explanations

In this section, we show that SME (with its enforced attribution to chemically meaningful fragments) allows one to more easily establish a deeper understanding on why certain fragments make the molecule more ideal for the target property. This deeper understanding is achieved by connecting fragments and some characteristic physiochemical properties that is conceptually known to be relevant to the property of interest, for instance, BBBP in this case.

Through the analysis of functional group attribution generated by SME, we can construct the desired relationship between molecular functional groups and the BBBP as shown in above **Figure 13A**. Similarly, by constructing different prediction models for fundamental physicochemical properties, we can construct the desired relationship between molecular functional groups and the fundamental physicochemical properties. Subsequently, we then establish the relationship between BBBP and the fundamental physicochemical properties, providing a complementary perspective on what molecular properties (substructure or physiochemical attributions) hold the key to having the desired properties. Four fundamental physicochemical properties include molecular weight (MW), topological polar surface area (TPSA), lipid solubility (LogP) and the number of hydrogen bond donor (HBDs) calculated by RDKit package are used as labels to construct different prediction models in this case. The overall process is shown in **Figure 13A** and the detailed result can be seen in **Figure 13 B-E**. As shown in **Figure 13 B-E**, SME does not find a clear correlation between BBBP and MW. Moreover, SME finds that BBBP is positively correlated with LogP and negatively correlated with HBDs and TPSA, which is consistent with the results reported in previous studies. The above results demonstrate that chemists can mine SAR information through SME and use this valuable information to further expose favorable

physicochemical properties these molecules must possess. Furthermore, converting information from favorable molecular substructures to favorable physicochemical/biochemical properties by SME is an abstraction that may actually inspire further understanding of complex phenomena as done by Mittal, A. et al.⁶⁵

Figure 13. (A). The flow chart of using SME to mine SAR information; (B). The correlation of the average attributions between BBBP and BBBP_MW's functional groups. (C). The correlation of the average attributions between BBBP and BBBP_LogP's functional groups. (D). The correlation of the average attributions between BBBP and BBBP_HBDs' functional groups. (E). The correlation of the average attributions between BBBP and BBBP_TPSA's functional groups.

Comment 4: How about sensitivity to the training data? There is some discussion on the consequence

of imbalanced data, but very little.

Reply:

We can think of two characteristics of a dataset that may impact the performance of SME: small dataset and highly imbalanced dataset. In this study, we note that SME has actually delivered satisfactory results on four different examples, all with relatively small data volumes: (ESOL (1111), BBBP (1859), hERG (9876) and Mutagenicity datasets (7672)), indicating that SME can be rather robust as long as the underlying GNN delivers a reasonable accuracy, such as a good F1 score on an imbalanced dataset. Of course, whether SME can uncover all essential SARs is questionable when the training dataset is way too small to give a reasonably representative picture of the prediction task at hand. In this case, we believe the underlying GNN predictive model is already not reliable, and some experts' judgments can easily expose potential problems: such as inconsistent interpretations (i.e. conflicting assignments of attribution scores to the same molecules fragments with high frequency) may be derived from SME. In short, SME's performance is directly tied to the 'coverage' of the training dataset. Potentially, a pre-trained GNN with other data sources could help alleviate the problem to some extent. Similarly, the issue with imbalance dataset if handled appropriately could also be partially mitigated. For instance, when training the model, we assign different weights (`pos_weight`) to the loss of positive samples to alleviate the problem of data imbalance.

Comment 5: *The possibility for replication of the work is well-served, with hyper-parameters clearly listed and with a Github repository. Still, the motivation for final hyper-parameter settings and choices are not well-explained.*

Reply:

Thanks for pointing this problem. The Tree Parzen Estimator (TPE) algorithm in Hyperopt is used for the hyperparameter optimization in this study. And we added the more detailed information about hyperparameters (The range of hyperparameters and the final optimized parameters) of RGCN in Table S3. Moreover, we added a description in section 2.3 (**Model Construction and Evaluation**) as follows:

“And the Tree Parzen Estimator (TPE) algorithm in Hyperopt (version 0.2.7) is used for the hyperparameter optimization in this study.”

Table S3 is as follows:

Table S3. The hyperparameters of different models.

Model	Parameters to be optimized	Package
ESOL	the number of nodes of each RGCN hidden layer: [64, 128, 256]	DGL 0.7.1
	the number of RGCN hidden layer: [2 , 3]	
	the number of nodes of each FC hidden layer: [64 , 128, 256]	
	the dropout rate of each RGCN hidden layer: [0, 0.1, 0.2, 0.3, 0.4, 0.5]	
	the dropout rate of each FC hidden layer: [0, 0.1 , 0.2, 0.3, 0.4, 0.5]	
	the learning rate: [0.003 , 0.001, 0.0003, 0.0001]	
	the number of epochs: 500	
	the patience of early stop: 30	
Mutagenicity	the number of nodes of each RGCN hidden layer: [64, 128, 256]	DGL 0.7.1
	the number of RGCN hidden layer: [2, 3]	
	the number of nodes of each FC hidden layer: [64, 128 , 256]	
	the dropout rate of each RGCN hidden layer: [0, 0.1, 0.2, 0.3, 0.4 , 0.5]	
	the dropout rate of each FC hidden layer: [0 , 0.1, 0.2, 0.3, 0.4, 0.5]	
	the learning rate: [0.003, 0.001 , 0.0003, 0.0001]	
	the number of epochs: 500	
	the patience of early stop: 30	
hERG	the number of nodes of each RGCN hidden layer: [64 , 128, 256]	DGL 0.7.1
	the number of RGCN hidden layer: [2, 3]	
	the number of nodes of each FC hidden layer: [64, 128 , 256]	
	the dropout rate of each RGCN hidden layer: [0, 0.1, 0.2 , 0.3, 0.4, 0.5]	
	the dropout rate of each FC hidden layer: [0, 0.1 , 0.2, 0.3, 0.4, 0.5]	
	the learning rate: [0.003, 0.001, 0.0003 , 0.0001]	
	the number of epochs: 500	
	the patience of early stop: 30	
BBBP	the number of nodes of each RGCN hidden layer: [64, 128, 256]	DGL 0.7.1
	the number of RGCN hidden layer: [2 , 3]	
	the number of nodes of each FC hidden layer: [64, 128 , 256]	

the dropout rate of each RGCN hidden layer: [0, 0.1, **0.2**, 0.3, 0.4, 0.5]

the dropout rate of each FC hidden layer: [0, 0.1, 0.2, 0.3, **0.4**, 0.5]

the learning rate: [0.003, **0.001**, 0.0003, 0.0001]

the number of epochs: 500

the patience of early stop: 30

Bold hyperparameters represent optimized hyperparameters.

Reviewer 3:

Key results

The presented approach derives chemically intuitive explanations for predictions of small molecules obtained from graph neural networks (and ensembles composed of such models). Deployed calculations facilitate a novel modification scheme of the attention layer, masking the contribution of meaningful substructures. Recorded difference between the prediction with the original and masked attention layer represent the feature importance. Utilized substructures can be freely chosen, but exemplary calculations for solubility, toxicity, and hERG activity were carried out using BRICS, Murcko scaffolds and the corresponding functional groups.

Comment 1:

Validity

Overall, conclusions drawn from the presented results are meaningful and correct, however I disagree that masking nodes reveals the importance of corresponding atoms, as after the message passing steps, nodes do not represent atoms, but molecular environments centered at the nodes. For example, masking a node originating from an amine group does not remove the information from a connected node, that it is adjacent to a nitrogen atom. Keeping this in mind, the explanations are still meaningful for the user, but it should be clarified in the text.

Reply:

We thank the reviewer for pointing out this very useful distinction. We fully agree with you on this point, masking a node can be considered as masking the molecular environments centered at the nodes (masking

main information of the central atom, and also masking part of the information of the adjacent atoms/bonds). For ease of understanding, in this article, the mask node is simply considered as masking the central atom.

And we have also discussed this perspective in a previous work (<https://doi.org/10.1021/acs.jmedchem.1c00421>). As shown in Figure A, we change the adjacent atom/bond, and then calculate the similarity of the node's (N atom here) embedding. The result demonstrated that the adjacent atom/bond will affect the embedding of the center node. And as the path grows, the affection of adjacent atoms/bond gradually weakens, which demonstrated that the chemical environment contains the information of adjacent atoms/bond, which gradually weakens as the path grows. Therefore, masking a node can be considered as masking the main information of the central atom, and also masking part of the information of the adjacent atoms/bonds.

Figure A. Similarity of the N's embedding (after message passing) with the changes in the surrounding substructures

As mentioned by the reviewer, the aforementioned observation does not affect the way we currently design SME, but these conceptual subtleties need to be clarified in the article. We added a description about this in the section 2.2.3. (**Substructure Mask**) as follows:

“After message passing, a node's hidden state gets updated by the information conveyed by adjacent atoms/bonds. Therefore, when we mask a node, it is not just about masking an atom but about masking

the chemical environment centered on that atom. And the chemical environment contains information of adjacent atoms, while the information of the adjacent atoms weakens as the path to the central atom grows. Therefore, masking a node can be considered as masking the main information of the central atom, and also masking part of the information of the adjacent atoms/bonds. For ease of understanding, in this article, the masking of a node is simply viewed as masking just the node itself.”

Comment 2:

Significance

Presented work adds value to the understanding of molecular predictions, however due to the abundance of existing and established explanatory approaches, it will not revolutionize the field of XAI, but represent a useful tool for the cheminformatics community.

The strength of proposed method also poses its greatest limitation: As chemically intuitive explanations are enforced, patterns learned from data-artifacts, which are intrinsically not chemically meaningful, may remain concealed. Hence SME, which excels in communicating results with non-ML-experts, and structure agnostic models for the validation of models are complementary in nature.

Reply:

We agree with the comment that “*As chemically intuitive explanations are enforced, patterns learned from data-artifacts, which are intrinsically not chemically meaningful, may remain concealed.*” This is indeed a possible limitation. Therefore, we also agree with the Reviewer’s comment that both approaches (chemically intuitive methods and structure agnostic methods) may be complementary. However, the structure agnostic methods dominate in the era of deep learning; we believe it is a good time to advocate chemically intuitive method like SME which enables chemists to appreciate more about deep-learning-based tools and to unleash their fullest potentials. Particularly, in this revised manuscript, we added one extra example on using SME to explain how a GNN predicts BBBP for molecules. We show that SME can even provide deeper understanding of how a model makes its prediction linking chemically meaningful substructures and physiochemical properties of molecules. Please see our response to the second part of comment 3 by Reviewer #2 for details.

In this sense, we argue the benefits of introducing SME to the chemists outweigh its potential drawbacks. Finally, while we repeatedly emphasize the benefits of SME in the manuscript, we did not want to dismiss existing structure agnostic methods. We thank the Reviewer to make this thoughtful

comment, and we have now made some clarifications in the Conclusion of the revised manuscript:

“Just like all other explanation methods for deep learning methods, SME also possesses several limitations. First, since SME (and other GNN-based interpretation methods) aims to explain what the GNN model has learned from the data, it is difficult for SME to mine reasonable SAR information when the GNN model has not learned the true or complete causality due to data paucity or data bias. Second, as chemically intuitive explanations are enforced, intrinsically chemically meaningless patterns learned from data artifacts may remain concealed. Lastly, the current version of SME only supports the three substructures of BRICS, Murcko and functional groups, and does not allow the assessment of some other substructures such as bioisosteres. Irrespective of these limitations (some can be easily overcome such as the last point mentioned above), SME provides intuitive and insightful interpretability for medicinal chemists based on chemically meaningful substructures. As most existing attribution methods for molecular GNNs in chemistry are directly borrowed from other fields and lacks the kind of chemical intuition that SME insists by design, we believe its benefits outweigh its drawbacks. Particularly, we demonstrate how SME help chemists in 4 key druggability properties tasks (ESOL, Mutagenicity, hERG and BBBP). In sum, SME is a tool with great potential to assist chemists to gain an appropriate understanding of why a GNN model makes certain prediction, and mine SAR information for structure optimization and de novo design.”

Comment 3:

Data and methodology

3.1 The overall methodology is valid, but as the authors already mentioned, explaining molecules split into multiple fragments may yield unexpected results. The rationale behind this issue is that calculating the importance as difference to the complete molecule violates criteria from the Shapley formalism, which ensure a fair attribution to each player (or feature). While the consequences are sufficiently discussed, it would be worth mentioning the origin of this issue.

Reply 3.1

Thanks for your suggestion. The Shapley value is assigned by beautiful and rigorous mathematical theory that the difference between the prediction and the average prediction is fairly distributed among the feature values (node or subgraph in GNN) of the instance. Since the graph data are complex, attribution calculation following the Shapley formalism are often time consuming. Furthermore, many works in

cheminformatics adopt an alternative approach for attribution: comparing the substructure differences between molecules in order to determine the attribution score of various substructures for the target property. This is a much faster and often satisfactory attribution method when there is only one dominant fragment responsible for the property of interest. This alternative attribution approach is not only widely adopted by traditional methods in cheminformatics but also often adopted to explain GNN models. Some representative works in this camp include: 1. Matched molecular pair analysis (MMPA) is a method in cheminformatics that compares the properties of two molecules that differ only by a single chemical transformation, such as the substitution of a hydrogen atom by a chlorine one. 2. Yurii Sushko et al combined the Matched molecular pair (MMP) and QSAR models to find MMP transformations based on QSAR predictions, and used the chemical interpretation of these transformations to give medicinal chemists useful hints for the hit-to-lead optimization process.⁶ 3. STONED-Counterfactuals (<https://pubs.rsc.org/en/content/articlehtml/2022/sc/d1sc05259d>) recommended by reviewer 1, it exploits STONED to enumerate many similar molecules and attempt to uncover ‘counter examples’ with respect to the given molecule of interest. Analyzing the SAR on these molecules allow chemists to arrive at an attribution of the key changed substructure in counter examples.

In view of the computational efficiency, SME is designed following this alternative approach. We choose to directly compare the difference between molecules and molecules with some substructures masked in order to determine the attribution. While this alternative approach will suffer issues like what we observed and commented in the paper, we also proposed a potential fix (in the original manuscript) that assigns a collective attribution score to combinations of fragments. Of course, the complexity of calculating attribution scores for fragment combinations quickly grows with system size. Hence, the idea of using Shapley values combining with our proposed fragmentation scheme in SME is a noteworthy approach. We thank Reviewer for bringing up this discussion, and we added the following comments (as advised) in the revised sec 2.2.4:

“For the sac of ensuring calculation efficiency, SME does not adopt Shapley values to fairly assign attribute values to different substructures, but only calculates attribute values based on the changes between predicted values, which may overlook contribution by some substructures. In order to solve the above problems, we adopt a compensation strategy: find the most positive/negative substructures through the combination of substructures to avoid overlooking some important substructures, and thus explore the SAR more comprehensively.”

3.2 Analytical approach & Suggested improvements

Statistical tests carried out in this study (correlation of average attribution and change of prediction) are meaningful and valid. However, the assessment of the explanation values should be carried out in a more systematic and objective manner (See suggested improvements).

Comparing atom highlighting with substructure highlighting, where each atom is colored according to the total contribution of the substructure overstates the clear contrast between positive and negative contributions, as the absolute values are greater and therefore more dominant in their coloring.

Reply 3.2:

In SME, we interpret the model based on the substructure, so SME highlights the entire substructure as the basic unit. This provides more intuitive information that is easier for chemists to understand. Our previous visualization method was not suitable enough, and it may be easy for people to misunderstand that we are still based on atoms, rather than looking at problems in fragments. Thank you very much for the visualization method you recommend. At the same time, using the visualization method you recommend can better understand our idea of taking fragments as the basic unit. Some examples are as follows:

3.3 (Also, attributions from the atom mask are assumingly not derived according to the Shapley formalism, and hence are limited in their interpretability.)

Reply 3.3:

As shown in the following figure, though the attributions from the atom mask are not derived according

to the Shapley formalism, this alternative approach (as stated earlier) is still widely accepted by the chemistry community and, empirically, enables chemists to establish useful SARs for explanations and structural optimizations. Let us briefly look at the example below. SME can capture and reflect which atoms are favorable for hydrophilicity. What we want to illustrate here is that the fragment-based interpretability of SME can provide a more intuitive and chemistry-compatible interpretation. Finally, we note that one may address the inadequacy of the atom-masking approach by deriving collective attribution scores that may expose further delicate interplay between the presence of various fragments in a molecule. However, we agree that Shapley formalism is definitely a very important and complementary approach. We have also more explicitly stated this possibility to replace the simplest atom-masking approach with Shapley if the computational cost is not a concern.

3.4 While this could be mentioned, a more subjective method to evaluate the explanatory power could be deployed, as presented by Rao et al. (<https://doi.org/10.48550/arXiv.2107.04119>). This would also address the more common use-case, where predictive performances of assessed models are mediocre.

Reply

Thanks for your suggestion. Quantitatively evaluating the explanatory power of interpretable models is

desirable, but it still faces enormous challenges. Rao et al. have made a good attempt in the fields and they defined a series of ground truths and used them to qualitatively evaluate the explanatory power of different interpretability methods. We have now cited the recommend work.

In our study, SME does not pursue the only absolute ground truth, but mining the corresponding SAR information by analyzing different molecular structure attribution. Moreover, SME is based on substructures while the quantitative evaluation methods provided by Rao's are based on atoms. Hence, Rao's method (as given) is not exactly compatible with SME. But it is indeed an alternative method that could work under the circumstances suggested by the Reviewer. We find Rao's work also discuss hERG cliffs, relevant to our work. Therefore, we added some hERG cliffs provided in the Rao's work as structural optimization examples in **Figure S1** to further validate SME. In the revised manuscript, we add the following comments and cite Rao's work:

"In addition, Rao et al. proposed a subjective method to evaluate the explanatory power of different interpretability methods and provide some hERG cliff molecular pairs. Some hERG cliff molecular pairs that did not appear in our training set also verify the effectiveness of SME in guiding structural optimization, and these results are shown in Figure S1."

Figure S1. Some real-world hERG cliff molecular pairs of hERG toxicity.

Comment 4: Colored substructure highlighting should not have colored atom labels.

Reply: Thanks for pointing this problem, and we remove the color of atom labels in the paper, and some examples are as follows.

Comment 5: Red/Green color-grading not accessible for visually impaired readers.

Reply: Thanks for pointing this problem, and we changed the color to a blue-red color scheme instead. In addition, our released codes provide options for different visual colors, and users can choose the color scheme they prefer according to their needs.

Comment 6: Distributions of attributions should be represented in a more established depiction, such as a box- or violin-plot.

Reply: Thanks for your valuable suggestion, and we change the figure to violin-plot, and the violin-plot makes the distributions of attributions clearer.

Comment 7: Ln 450. [...] 0.003 to the amino group.

Reply: Thank you and we already correct this minor mistake.

Comment 8: Ln 492. You mean the mutagenicity dataset?

Reply: We thank the reviewer and have corrected the mistake.

Comment 9: Ln 625/646/653. Figure 12x?

Reply: We thank the reviewer and have corrected the mistake.

Comment 10:

Optional:

Substructure highlighting may benefit from using code presented at:

<http://rdkit.blogspot.com/2020/10/molecule-highlighting-and-r-group.html>

Reply: Thanks for your valuable suggestion, and we used the substructure highlighting methods you recommend above, and it makes our visualizations more intuitive. Once again, we thank you again for your constructive comments on substructure highlighting and figures, and we expressed in acknowledgements as follows:

“The authors are grateful to the anonymous reviewer for his/her helpful suggestions for substructure highlighting and figures in this paper.”

Comment 11:

Optional:

Clarity and context

Your view on the clarity and accessibility of the text, and whether the results have been provided with sufficient context and consideration of previous work. Note that we are not asking for you to comment on language issues such as spelling or grammatical mistakes.

The manuscript is well written and easy to access.

References

Your view on whether the manuscript references previous literature appropriately.

Ref 24: The only source I found (openreview.net) marked this work as rejected. It should be removed.

In the listing of explanatory approaches “Improving Molecular Graph Neural Network Explainability with Orthonormalization and Induced Sparsity” from Henderson et al. could be mentioned.

Reply: Thanks for pointing this problem, and we remove the Ref 24. In addition, “Improving Molecular Graph Neural Network Explainability with Orthonormalization and Induced Sparsity” is an excellent work and we cite it.

Reference

- 1 Mobley, D. L. & Guthrie, J. P. FreeSolv: a database of experimental and calculated hydration free energies, with input files. *Journal of computer-aided molecular design* **28**, 711-720 (2014).
- 2 Xu, C. *et al.* In silico prediction of chemical Ames mutagenicity. *Journal of chemical information and modeling* **52**, 2840-2847 (2012).
- 3 Zhang, C. *et al.* In silico prediction of hERG potassium channel blockage by chemical category approaches. *Toxicology research* **5**, 570-582 (2016).
- 4 Xiong, G. *et al.* ADMETlab 2.0: an integrated online platform for accurate and comprehensive predictions of ADMET properties. *Nucleic Acids Research* **49**, W5-W14 (2021).
- 5 Polishchuk, P. G., Kuz'min, V. E., Artemenko, A. G. & Muratov, E. N. Universal approach for structural interpretation of QSAR/QSPR models. *Molecular Informatics* **32**, 843-853 (2013).
- 6 Sushko, Y. *et al.* Prediction-driven matched molecular pairs to interpret QSARs and aid the molecular optimization process. *Journal of Cheminformatics* **6**, 1-18 (2014).

REVIEWERS' COMMENTS

Reviewer #1 (Remarks to the Author):

In general all my comments are satisfied. One minor comment is the treatment of tautomers. I think they are removed from the datasets and may not work well with the method. I think this should be made more explicit - clearly tautomers are trouble with GNNs and not anything unique to the interpretation - but it would be good to mention as a challenge for the future.

Reviewer #2 (Remarks to the Author):

Thank you very much for very clear notation and explanations of your updates to this work. I appreciate the work done in considering my remarks on the submitted paper:

An extensive discussion about validity and generalizability of the approach was developed. An additional experiment, for blood-brain barrier permeation, was added that complements the three specific molecular (druggability) properties (solubility, mutagenicity and hERG toxicity).

The argumentation about generic chemists' understanding is elaborated and argues well about the value of the proposed method. I agree that full validation of generic chemists' understanding is very difficult, but I also think that some benchmarking would be useful. It is however not reasonable to develop that within the work of this paper, I think.

The rebuttal text elaborates well on the hyperparameter selections made.

The Conclusions section has been expanded to cover a thorough discussion covering possible limitations of the approach. The arguments are well elaborated and convincing. This discussion further mitigates common issues for criticism of studies like this, in both the machine learning and cheminformatics areas, I think.

I have no further remarks on this publication.

Reviewer #3 (Remarks to the Author):

The revised manuscript as well as the comments of the author fulfill my expectations.

Reviewer 1:

Comments:

In general all my comments are satisfied. One minor comment is the treatment of tautomers. I think they are removed from the datasets and may not work well with the method. I think this should be made more explicit - clearly tautomers are trouble with GNNs and not anything unique to the interpretation - but it would be good to mention as a challenge for the future.

Reply:

We thank the reviewer for encouraging and helpful comments. The absence of tautomers structures in the training set can impact the model's prediction accuracy, thus hindering the application of the SME method. This aligns with our first identified limitation, as SME can't mine reasonable SAR information when the underlying GNN model has not learned the true or complete causality due to data paucity or bias. Therefore, we have added the following remark in the article. *“As a noteworthy example, tautomers present a challenge in the pursuit of accurate molecular property predictions and, accordingly, has implications for the effectiveness of SME.”*